# Contributions to local and regional-scale formaldehyde concentrations

Lucas A. J. Bastien[1,2], Nancy J. Brown[2], and Robert A. Harley[1,2]

[1]Department of Civil and Environmental Engineering, University of California at Berkeley, Berkeley CA 94720
[2]Energy Analysis and Environmental Impacts Division, Lawrence Berkeley National Laboratory, 1 Cyclotron Road, Berkeley CA 94720

**Correspondence:** Robert A. Harley (harley@ce.berkeley.edu)

**Abstract.** Reducing ambient formaldehyde concentrations is a complex task because formaldehyde is both a primary and a secondary air pollutant, with significant anthropogenic and biogenic sources of volatile organic compounds (VOC) precursor emissions. This work uses adjoint sensitivity analysis in a chemical transport model to identify emission sources and chemical reactions that influence formaldehyde mixing ratios in the San Francisco Bay Area, and within three urbanized sub-areas. For each of these receptors, the use of the adjoint technique allows for efficient calculation of the sensitivity of formaldehyde to emissions of $NO_x$, formaldehyde, and VOC precursors occurring at any location and time. Formaldehyde mixing ratios are found to be generally higher in summer than in winter. The opposite seasonal trend is observed for the sensitivities of these mixing ratios to formaldehyde emissions. In other words, even though formaldehyde is higher in summer, reducing formaldehyde emissions has a greater impact in winter. In winter, 85–90 % of the sensitivity to emissions is attributed to direct formaldehyde emissions. In summer, this contribution is smaller and more variable, ranging from 27 to 75 % among the receptor areas investigated in this study. Higher relative contributions of secondary formation versus direct emissions are associated with receptors located farther away from heavily urbanized and emission-rich areas. In particular, the relative contribution of biogenic VOC emissions (15–43 % in summer) is largest for these receptors. Ethene and other alkenes are the most influential anthropogenic precursors to secondary formaldehyde. Isoprene is the most influential biogenic precursor. Sensitivities of formaldehyde to $NO_x$ emissions are generally negative, but relatively small in magnitude compared to sensitivities to VOC emissions. The magnitude of anthropogenic emissions of organic compounds other than formaldehyde is found to correlate reasonably well with their influence on population-weighted formaldehyde mixing ratios at the air basin scale. This correlation does not hold for ambient formaldehyde in smaller urbanized sub-areas. The magnitude of biogenic emissions does not correlate with their influence in either case.

# 1 Introduction

Formaldehyde is one of the major contributors to increased cancer risk associated with exposure to hazardous air pollutants in the United States (USEPA, 2012; Strum and Scheffe, 2016). Based on observations, Strum and Scheffe (2016) estimated that the cancer risk associated with exposure to ambient formaldehyde in major U.S. cities was of the order of 50 per million people, with values reaching 150 per million people in the southeast part of the country. In contrast, Zhu et al. (2017) estimated using a modeling approach that the upper bound for this cancer risk was in the range 20–40 per million people. Average ambient formaldehyde mixing ratios in U.S. urban areas typically range from 1 to 3 ppb (Strum and Scheffe, 2016), where 1 ppb = 1 nmol/mol. Peak mixing ratios can be an order of magnitude larger (Dasgupta et al., 2005). In contrast, background mixing ratios are estimated to be in the range 0.1 to 0.2 ppb (McCarthy et al., 2006).

Formaldehyde is directly emitted by anthropogenic activities that include a number of industrial processes and combustion of fossil fuels in stationary applications and motor vehicles (e.g. Grosjean et al., 2001; Ban-Weiss et al., 2008). Formaldehyde is also formed photochemically in the atmosphere through oxidation of other volatile organic compounds (Finlayson-Pitts and Pitts Jr., 2000; Seinfeld and Pandis, 2016). Examples of chemical pathways that can lead to secondary formation of formaldehyde are: addition of OH, ozone, or the nitrate radical to the C=C double bond of alkenes, and photolysis of acetaldehyde. In the presence of sunlight, formaldehyde undergoes photolysis following one of two pathways (Seinfeld and Pandis, 2016):

$$HCHO + h\nu \rightarrow H^\bullet + HCO^\bullet \overset{2O_2}{\rightarrow} 2HO_2^\bullet + CO \tag{R1}$$

or:

$$HCHO + h\nu \rightarrow H_2 + CO \tag{R2}$$

where $h$ is Plank's constant and $h\nu$ is the energy of a photon of frequency $\nu$. In addition to photolysis, removal of formaldehyde occurs through abstraction of a hydrogen atom by OH radicals, and to a lesser extent by free chlorine atoms and nitrate radicals. These reactions form formyl radicals and, respectively, water, hydrochloric acid, and nitric acid (Finlayson-Pitts and Pitts Jr., 2000; Seinfeld and Pandis, 2016). For example with OH:

$$HCHO + OH^\bullet \rightarrow H_2O + HCO^\bullet \tag{R3}$$

As indicated in the reactions presented above, formaldehyde is a source of radicals, and therefore affects the formation and removal of other air pollutants such as ozone and secondary organic aerosols (Xiaoyan et al., 2010; Luecken et al., 2012; Zheng et al., 2013; Seinfeld and Pandis, 2016). While other carbonyl species undergo photolysis and are also sources of radicals, the photolysis rate coefficient and therefore the potential for radical production are much larger for formaldehyde (Finlayson-Pitts and Pitts Jr., 2000).

Mitigating exposure to ambient formaldehyde requires a clear understanding of the contributions of primary versus secondary and anthropogenic versus biogenic sources of formaldehyde. Various approaches have been used to estimate the relative

importance of direct emissions versus secondary formation of formaldehyde. One approach consists of studying the statistical relationships between observed mixing ratios or concentrations of formaldehyde and species that are predominantly of primary origin (e.g. CO) versus species that are predominantly of secondary origin (e.g. $O_3$). Using this approach, the average contribution of primary emissions to total ambient formaldehyde in Houston, TX was estimated to be a third to a half (Friedfeld et al.,
2002; Rappenglück et al., 2010); similar findings are reported for Mexico City (Garcia et al., 2006). Parrish et al. (2012) discuss the limitations and sources of uncertainties inherent to these statistical methods. Using an emission inventory constrained by observations and product yields of formaldehyde from oxidation of some of its precursors, they attributed over 90 % of ambient formaldehyde over the Houston region to secondary formation. Regardless of the method being used, results derived for one location may not be directly applicable to another because several factors, including emissions and meteorology, can
influence secondary formation of formaldehyde and of other species. For example, many petrochemical industries are located in Houston and its surroundings. Emissions from this area may be particularly different from those of other large cities.

The contribution of direct emissions versus secondary formation of aldehydes has also been investigated using modeling approaches. Luecken et al. (2012) used a chemical transport model where species that form formaldehyde are tracked and allocated to the emitted precursor species from which they originated. These investigators estimated the relative contribution
of secondary formation to total ambient formaldehyde in the eastern U.S to be 70–90 % in July versus 30–90 % in January, with significant spatial variability. They also found that alkenes were often the most significant precursors to formaldehyde, with isoprene contributing 15–60 % of secondary formation in certain areas in July. Uncertainties in emission inventories, particularly for alkenes, an incomplete understanding of atmospheric reactions that influence formaldehyde concentrations, and the condensation of chemical mechanisms in air quality models have been proposed as likely sources of uncertainties in model
simulations (Luecken et al., 2012, 2018). Additionally, formaldehyde concentrations exhibit sharp spatial gradients at sub-kilometer scales (Kheirbek et al., 2012), further complicating modeling efforts, as horizontal grid sizes in model simulations typically range from 4 to 32 km in regional air quality models (Fountoukis et al., 2013).

Sensitivity analysis is another modeling approach used to investigate the drivers of air pollution. The aim is to quantify the effects of model parameters such as emission sources on air quality (Cohan and Napelenok, 2011). Most sensitivity analysis
techniques can be classified as either source-oriented or receptor-oriented. Source-oriented techniques, such as the Decoupled Direct Method (DDM, Dunker, 1981, 1984; Yang et al., 1997; Dunker et al., 2002; Hakami et al., 2003; Napelenok et al., 2006; Koo et al., 2007), require one large-scale simulation for each parameter of interest, but allow for quantifying sensitivities of multiple air quality metrics at little additional cost. In contrast, receptor-oriented techniques, such as the adjoint method (Sandu et al., 2005; Martien et al., 2006; Hakami et al., 2007; Henze et al., 2007; Bastien et al., 2015), require one large-scale
simulation for each air quality metric of interest, but allow for the efficient calculation of sensitivities to large numbers of model parameters. While there have been numerous analyses of ozone and particulate matter, little attention has been devoted to analysis of formaldehyde mixing ratios. Dunker et al. (2015) used the path-integral method with sensitivities calculated with the DDM, and found that the relative participation of different anthropogenic emission sectors (e.g. point sources, light duty vehicles) in the overall reduction of formaldehyde depended on the order in which these emissions were removed. The
path-integral method (Dunker, 2015) uses sensitivity coefficients to apportion changes in mixing ratios associated with specific

emission changes. More recently, Luecken et al. (2018) used the DDM to apportion U.S. ambient formaldehyde to its precursor emissions. They estimated the relative contribution of secondary formation to be in the range 60–90 %, with alkenes being the most influential precursors.

Our understanding of the relative contributions to formaldehyde exposure of primary versus secondary formation and of biogenic versus anthropogenic emissions is still incomplete. In particular, we have yet to fully characterize how this apportionment varies across sub-areas of an urbanized region. The present study uses a chemical transport model and its adjoint to identify the emission sources and chemical reactions that influence ambient formaldehyde, overall for the San Francisco Bay Area, and within three urbanized sub-areas. The relative influence of primary versus secondary formation, anthropogenic versus biogenic emissions, and of different anthropogenic emission sectors is quantified for each urban sub-area and for the entire air basin. The sensitivities of formaldehyde to rate coefficients of over 200 chemical reactions are calculated and ranked. The study is conducted for two month-long simulation periods, in July and December, to study the seasonality effects on the analysis. The use of the adjoint approach allows for spatial, temporal, and by-species resolution in the calculation of sensitivities at a reasonable computational cost, such that reaction rate coefficients and emissions of each species at each location and at each hour of the simulations can be treated as distinct model parameters. There are billions of such sensitivities, and the computational cost of this endeavor would be prohibitive if a source-oriented sensitivity analysis technique were used in place of the adjoint method. The approach employed here can be used, for example, to draw "maps of influence"; these identify upwind areas whose emissions have the greatest influence on model responses. To the best of our knowledge, this study is the first to use adjoint sensitivity analysis to identify the emissions and chemical reactions that have the most influence on regional- and local-scale ambient formaldehyde.

## 2   Methods

### 2.1   Chemical transport modeling, air quality metrics, and sensitivity analysis

Chemical transport models (CTMs) are used to synthesize understanding of atmospheric processes relevant to air pollution, test new scientific hypotheses, estimate pollutant levels where there are no available observations, and estimate past or future changes in air quality due, for example, to emission reductions or climate change. CTMs estimate mixing ratios of airborne pollutants of interest over a user-defined gridded domain and simulation period, by numerically solving a system of coupled, and in general non-linear, partial differential equations that express conservation of mass (Seinfeld and Pandis, 2016). Model input parameters include: meteorology, emissions of pollutants of interest and their precursors, description of relevant atmospheric chemical reactions, mixing ratios at the beginning of the simulation period (initial conditions) and pollutant levels in air entering the modeling domain through lateral boundaries and from above (boundary conditions).

CTMs estimate space- and time-resolved fields of pollutant mixing ratios, but typically do not provide information about which model parameters (e.g. which emission sources and which chemical reactions) contribute to specific air pollution problems. Mathematical and numerical tools that aim at elucidating these relationships have been implemented in CTMs. In particular, sensitivity analysis in air quality models quantifies changes in a specified air quality metric $R$ resulting from variations in

model parameters $\alpha_1, \alpha_2, \ldots, \alpha_N$ through the calculation of first-order sensitivity coefficients of the form:

$$S_{\alpha_i}^+ = \frac{\partial R}{\partial \alpha_i} \tag{1}$$

or, in semi-normalized form:

$$S_{\alpha_i}^\times = \frac{\partial R}{\partial \alpha_i} \alpha_i \tag{2}$$

Air quality metrics of interest are calculated from model output mixing ratios. This study uses sensitivity analysis to identify emission sources and chemical reactions that influence metrics related to exposure to ambient formaldehyde. More precisely, the metrics of interest are defined as population-weighted formaldehyde mixing ratios of the form:

$$R = \int\limits_\omega \int\limits_T P_\omega(x_1, x_2) C(x_1, x_2, t) dt dx_1 dx_2 \tag{3}$$

where $\omega$ is the receptor area of interest, $T$ is the simulation period excluding spin-up, $P_\omega$ is the population-weighting probability

density function over receptor $\omega$, $C$ is the surface mixing ratio of formaldehyde, and $(x_1, x_2)$ and $t$ are the horizontal position and time coordinates, respectively. Population-weighting functions are based on 2010 U.S. census data. Four receptor areas are investigated (Fig. 1). Three of these receptor areas (Pittsburg/Antioch, San Jose, and the East Bay) are urban areas that have been identified by the Bay Area Air Quality Management District as heavily impacted by air pollution (BAAQMD, 2014). The fourth receptor is defined as the entire San Francisco Bay Area air basin. Each model response $R$ is a real-valued function that

depends on the model parameters $\alpha_1, \alpha_2, \ldots, \alpha_N$: $R = R(\alpha) = R(\alpha_1, \alpha_2, \ldots, \alpha_N)$. Model parameters are denoted collectively as a vector $\alpha = (\alpha_1, \alpha_2, \ldots, \alpha_N)$, where $N$ is the total number of parameters.

   Sensitivity analysis is used to calculate and compare changes $\Delta R$ in model response resulting from different perturbations in model parameters at any location in the domain and any time in the simulation period. This study focuses on two categories of parameters: emissions and reaction rate coefficients. Changes in parameters are calculated with respect to nominal values,

denoted $\alpha^0 = (\alpha_1^0, \alpha_2^0, \ldots, \alpha_N^0)$. Nominal values are meant to describe actual conditions of the reference year. Assuming that $R$ is differentiable at $\alpha^0$, the change $\Delta R$ in model response corresponding to a change in model parameters from nominal values $\alpha^0$ to arbitrary values $\alpha$ is given by:

$$\Delta R = R(\alpha) - R(\alpha^0) \tag{4}$$

$$= \sum_{i=1}^N \left. \frac{\partial R}{\partial \alpha_i} \right|_{(\alpha^0)} \Delta \alpha_i + h_{\alpha^0}(\alpha) \tag{5}$$

where $\Delta \alpha_i = (\alpha_i - \alpha_i^0)$ is the change in parameter $\alpha_i$ and $h_{\alpha^0}$ is a real-valued function (a priori a different function for each set of nominal values $\alpha^0$) with domain in $\mathbb{R}^N$, and that has the property:

$$\lim_{\substack{\Delta \alpha_i \to 0 \\ i=\{1,2,\ldots,N\}}} h_{\alpha^0}(\alpha) = 0 \tag{6}$$

Provided that $R$ is sufficiently differentiable at $\alpha^0$, the value of $h_{\alpha^0}(\alpha)$ can be calculated from the knowledge of higher-order derivatives of the form $\partial^\beta R / (\partial \alpha_1^{\beta_1} \partial \alpha_2^{\beta_2} \cdots \partial \alpha_N^{\beta_N})$ evaluated at $\alpha^0$, where $\beta = \beta_1 + \beta_2 + \cdots + \beta_N \geqslant 2$. This study focuses on

first-order sensitivities and makes the assumption that model responses vary linearly with model parameters, in which case Equation 5 simplifies to:

$$\Delta R = \sum_{i=1}^{N} \frac{\partial R}{\partial \alpha_i}\bigg|_{(\alpha^0)} \Delta \alpha_i = \sum_{i=1}^{N} S_{\alpha_i}^{+} \Delta \alpha_i \tag{7}$$

The value of $h_{\alpha^0}(\alpha)$ measures the error made when using Equation 7 in lieu of Equation 5. As indicated by Equation 6, the linear approximation holds exactly for infinitesimally small perturbations but not necessarily for larger perturbations. Previous modeling studies investigated the range of perturbations over which Equation 7 holds reasonably well. Vuilleumier et al. (1997) used a Green's function approach in a box model to calculate first- and second-order sensitivities of ozone to initial concentrations of precursors and to reaction rate coefficients. They found that using first-order sensitivities to estimate changes in modeled ozone resulting from perturbing model parameters yielded acceptable results for perturbations of at most 25 % of the baseline values. Hakami et al. (2003, 2004) found good agreement ($r^2 \geqslant 0.99$) between first-order sensitivities of ozone to domain-wide NO and VOC emissions calculated using the DDM on the one-hand and a central finite-difference approximation with $\pm$ 10 % perturbations on the other hand. Additionally, Hakami et al. found that modeled ozone exhibited more nonlinear behavior when (i) perturbing $NO_x$ emissions compared to perturbing VOC emissions; (ii) ozone concentrations were high; and (iii) ozone chemistry was in the transition regime between VOC- and $NO_x$-controlled conditions. Cohan et al. (2005) found that changes in modeled ozone resulting from 10 % perturbations in domain-wide $NO_x$ and VOC emissions were accurately estimated using a Taylor series truncated after the first-order term. Overall, these studies suggest that ozone responses vary nearly linearly with emissions of precursor species for perturbations up to about 10 % of the nominal value, and up to 25 % in some cases. Since formaldehyde is a ubiquitous intermediate species in atmospheric oxidation cycles, we hypothesize that formaldehyde responses also vary approximately linearly with precursor emissions for such perturbations. Using the sensitivity results presented in this study for large (> 25 %) perturbations that are likely to result in nonlinear behavior should be avoided. In particular, non-linear behavior is likely to be significant when first-order sensitivities are large, and in regions where ozone chemistry is near the transition regime between VOC- and $NO_x$-controlled conditions (Vuilleumier et al., 1997; Hakami et al., 2003, 2004; Cohan et al., 2005; Jin et al., 2008).

Both the first-order sensitivity $S_{\alpha_i}^{+}$ and its semi-normalized counterpart $S_{\alpha_i}^{\times}$ are useful in quantifying the sensitivity of air quality metrics to model parameters, but these quantities differ in their interpretation. $S_{\alpha_i}^{+}$ measures the change $\Delta R$ in metric $R$ resulting from increasing the value of parameter $\alpha_i$ by an amount $\Delta \alpha_i$, whereas $S_{\alpha_i}^{\times}$ measures the change $\Delta R$ in metric $R$ resulting from scaling parameter $\alpha_i$ using a multiplicative factor (e.g. $\xi = 0.1$ for a 10 % increase in $\alpha_i$):

$$\Delta R \approx S_{\alpha_i}^{+} \Delta \alpha_i \tag{8}$$

$$\Delta R \approx S_{\alpha_i}^{\times} \xi \tag{9}$$

Semi-normalized sensitivities are zero whenever the underlying parameter is itself zero. In contrast, this statement does not hold in the general case for $S_{\alpha_i}^{+}$. All model parameters of interest in this study vary in time and space, and the values of these parameters at each location and at each time step are regarded as distinct parameters. Sensitivities calculated here are therefore local quantities. For example, the semi-normalized sensitivity of a model response $R$ to the formaldehyde emission rate

$E_{\mathrm{HCHO}}(\boldsymbol{x},t)$ at location $\boldsymbol{x}$ and time $t$ is $S^{\times}_{E_{\mathrm{HCHO}}(\boldsymbol{x},t)}$. Semi-normalized sensitivities may be grouped together. For example, the semi-normalized sensitivity to VOC emissions is the sum of the semi-normalized sensitivities to emissions of each individual species that is used to represent VOC in the model.

## 2.2 Computational approach

The Community Multiscale Air Quality model (CMAQ, Byun and Schere, 2006) is used at 4 km horizontal resolution to simulate atmospheric formaldehyde mixing ratios in the San Francisco Bay Area, during two month-long simulation periods: summer (July) and winter (December). The modeling domain consists of $38 \times 52$ grid cells in the horizontal directions and 35 vertical layers, extending to about 15 km above ground level. Atmospheric photochemistry is simulated with the SAPRC99 chemical mechanism (Carter, 2000). SAPRC99 comprises 72 model species and 211 chemical reactions (listed in Carter, 2000,

Appendix A). Some model species represent individual chemical species such as formaldehyde, ozone, ethene, and isoprene. Other model species lump multiple organic species together, based on similarities in molecular structures and/or reactivity. Lumped model species include ALK1 through ALK5 ("alkanes and other non-aromatic compounds that react only with OH", Carter, 2000), OLE1 and OLE2 (alkenes other than ethene), ARO1 and ARO2 (aromatics), and TERP (terpenes). The numbers in lumped model species names for a given molecular structure (e.g. OLE1 versus OLE2) separate species according to their

reactivity with the OH radical, a higher number meaning higher reactivity (Carter, 2000).

Hourly gridded emissions at 4 km horizontal resolution were provided by the Bay Area Air Quality Management District (BAAQMD, 2009) for the year 2005. The emissions data account for weekday–weekend and seasonal variations in anthropogenic and biogenic emissions. Meteorological conditions were modeled at 4 km resolution using the MM5 model (BAAQMD, 2009). A study by Bastien et al. (2015) suggests that year 2000 meteorological conditions used here represent

typical conditions for multi-week simulation periods. Initial and boundary conditions for 34 out of the 72 model species described in the SAPRC99 chemical mechanism were derived from the global-scale MOZART-4 modeling system (Emmons et al., 2010). Boundary conditions for these species vary in the vertical direction, and also vary hourly throughout the simulation. The remaining 38 SAPRC99 model species could not be mapped to any MOZART-4 species. Initial and boundary conditions for these species were set to zero. A two-day model spin-up period is used to reduce the influence of uncertainties

in initial conditions on this analysis.

CMAQ is used to calculate model responses of interest for nominal values of the model parameters. The adjoint of CMAQ is used to calculate the local sensitivity coefficients $S^{+}_{\alpha_i}$ and $S^{\times}_{\alpha_i}$. The use of adjoint methods to calculate first-order sensitivity coefficients has been described elsewhere (Sandu et al., 2005; Martien et al., 2006; Hakami et al., 2007; Henze et al., 2007; Bastien et al., 2015). As previously explained, the adjoint sensitivity technique is particularly attractive when the number of

30 model parameters of interest is large compared to the number of model responses of interest. Eight model responses are investigated here (2 seasons × 4 receptor areas) but there are billions of model parameters: emissions and reaction rate coefficients at each location and each hourly time step. The computer code used in this study is based on the work by Hakami et al. (2007) with modifications described in Bastien et al. (2015). Additionally, the CB-IV chemical mechanism used in Hakami et al. (2007) was replaced by the SAPRC99 chemical mechanism. The chemical solver and adjoint sensitivity routines for

SAPRC99 were created with the Kinetic Pre-Processor (KPP, Sandu et al., 2003; Daescu et al., 2003; Sandu and Sander, 2006) version 2.2.3. The original adjoint code was further adapted so that it could be run using multiple processors in a parallel computing environment. This study expands on previous work of Bastien et al. (2015), where emission sources that influenced mixing ratios of two primary air pollutants (benzene and diesel black carbon) were mapped. Benzene and diesel black carbon were treated as non-reactive, whereas in the present study formaldehyde is reactive and is also formed photochemically from other VOC precursors.

## 3 Results and discussion

### 3.1 Model evaluation

The forward model is the component of the model that estimates ambient mixing ratios of model species for a given set of input parameters. In other words, the forward model relates an emission inventory and a set of meteorological conditions to the resulting air quality in the modeling domain, accounting for atmospheric chemistry and other relevant processes. In contrast, the adjoint of the forward model is the component of the model used to calculate first-order sensitivities i.e. rates of change of air quality when input parameters such as emissions are perturbed. Once the forward model has been run for a given set of input parameters, the adjoint model can be run once for each air quality metric of interest without re-running the forward model. The accuracy of these two components are evaluated separately.

The forward model is evaluated by comparing modeled mixing ratios of a number of species to ground-based observations from routine monitoring stations (Fig. S1 through S16 and Table S1). Hourly measurements of $O_3$ were available at 20 sites, hourly measurements of NO, $NO_2$, and CO were available at 13 sites, and daily observations (24-hour averages) of formaldehyde, acetaldehyde, and acetone were available at 3 sites. The model captures the magnitude and diurnal variations of summer-time $O_3$ mixing ratios at a majority of the sites. The corresponding mean bias ranges from $-4.4$ to $10.5$ ppb and is within $\pm 5$ ppb at 12 out of the 20 locations where observations are available. The corresponding root mean square error (RMSE) ranges from 3.5 to 12.7 ppb. Comparing average daily profiles of $O_3$ + $NO_2$ mixing ratios yields mean bias in the range $-2.3$–$7.0$ ppb and RMSE in the range 2.3–7.6 ppb. Model performance is weaker for summer season NO, $NO_2$, and $NO_x$ = NO + $NO_2$ when compared to results for $O_3$ and $O_3$ + $NO_2$. $NO_x$ is sometimes dramatically overestimated. In contrast, CO is consistently underestimated by the model. This issue may be a consequence of the size of model grid cells: in the model, local emissions are instantaneously diluted within a 4 km x 4 km grid cell. This artificial dilution has more influence on pollutants that are directly emitted, such as CO and NO, than on secondary pollutants such as ozone.

Model performance is less satisfactory in winter for ozone, oxides of nitrogen, and carbon monoxide (Table S1), which suggests that winter-time atmospheric chemistry is not as well represented by the model. However, results presented in the following sections indicate that atmospheric chemistry has little influence on formaldehyde concentrations in winter. In this case, formaldehyde behaves mostly as a non-reactive tracer species. The performance of the model to predict tracer species concentrations in both summer and winter has previously been evaluated by Bastien et al. (2015). Here, the model evaluation includes comparisons of modeled versus observed daily-averaged mixing ratios of formaldehyde, acetaldehyde, and acetone

(Fig. S14 through S16). Mean modeled mixing ratios of formaldehyde are within $\pm\,26\,\%$ of observed values at all sites except Arkansas Street in San Francisco in July (relative difference = 34 %). There is no consistent trend (overestimation versus underestimation) in these differences. In contrast, mean mixing ratios of acetaldehyde are consistently underestimated by the model, but are within 34 % of observed values at all sites except Chapel Way in Fremont in July (relative difference = $-55$ %). The model consistently underestimates acetone mixing ratios by 56 to 81 %. A possible explanation for the underpredictions of acetone is that emissions may be significantly underestimated (McDonald et al., 2018).

The adjoint of the forward model is evaluated by comparing the change $\Delta R$ in model response estimated using adjoint sensitivities versus values calculated using a brute-force approach, which involves running the forward model with perturbed parameters and comparing the outputs with those from the baseline simulation. Since the calculation of each brute-force estimate requires a separate model run, this evaluation is conducted for one-day episodes in July and December, in each case the first day of the corresponding month-long simulation period. The model responses used for evaluation are domain-wide population-weighted mixing ratios of formaldehyde and ozone. The ozone responses are defined similarly to the formaldehyde responses (i.e. Eq. 3). Additive 24-hour perturbations to emissions of NO, formaldehyde, ethene, and isoprene at several locations within the modeling domain are used for the evaluation. The magnitude of the perturbations for a given species is set to be roughly 10 % of the maximum emission rate observed at any location of the modeling domain and throughout the simulation period. Adjoint and brute-force estimates of $\Delta R$ are in good agreement for perturbations in formaldehyde, ethene, and isoprene emissions (Fig. 2), with a corresponding coefficient of determination $r^2 = 0.95$ or above. The agreement is less satisfactory for perturbations in NO emissions, and including the corresponding data points in the analysis lowers the coefficient of determination to the 0.86–0.93 range. Discrepancies between adjoint-based and brute-force estimates are generally larger when $\Delta R$ is large.

The good agreement between brute-force and adjoint sensitivities of formaldehyde to emissions of precursors other than NO supports the hypothesis that formaldehyde responses vary nearly linearly with these emissions for perturbations on the order of 10 % of baseline values; the brute-force approach includes non-linearities whereas first-order adjoint sensitivities do not. This hypothesis is likely to hold for somewhat larger perturbations since the perturbations used for creating Fig. 2 were in most cases larger than 10 % of baseline values. A likely explanation for the larger mismatch for NO shown in Fig. 2 is that non-linearities in the relationship between model responses and emissions are stronger for $NO_x$ than for emissions of other species.

## 3.2 Sensitivity apportionment

This section identifies species whose emissions most strongly influence formaldehyde model responses. All the sensitivity values presented and discussed in this section are "overall sensitivities", obtained by summing individual semi-normalized sensitivities over the modeling domain and the month-long duration of the simulation. The next section discusses the influence of the location of sources relative to receptors on model responses. It is worth noting that summing non-normalized sensitivities over the modeling domain would measure the air quality effects of adding a fixed amount of emissions at all locations, including over the ocean, where there are few anthropogenic sources. The summing procedure is only applied to semi-normalized

sensitivities, and the overall sensitivity of a model response to emissions of species $i$ measures the effects on that response of uniformly scaling up the existing emissions of this species. Sensitivities to initial conditions of all model species were also calculated, but were found to be negligible compared to emissions sensitivities, indicating that the spin-up and simulation periods are long enough such that initial conditions have negligible influence on model responses of interest here.

The magnitude of population-weighted formaldehyde mixing ratios (i.e. the model responses, Table 1) is in the 1.3–1.5 ppb range for all receptor areas in July, decreasing by 13 to 61 % going from July to December. San Jose sees the smallest seasonal decrease and Pittsburg/Antioch has the largest decrease. In contrast, mixing ratios of many formaldehyde precursors are higher in December than in July (Tables S2 through S5).

     Figures 3a and 3b show the emissions sensitivity of the air basin-wide response for summer (July) and winter (December),
respectively. Organic species other than methane are grouped together as VOC, and likewise NO and $NO_2$ are grouped together as $NO_x$. Of the remaining species, those whose relative contribution is smaller than 5 % are lumped together as "others" in the gray portions of each subplot, separately for positive and negative sensitivities. The sensitivity to VOC emissions is further separated into the contributions of different source sectors: biogenic emissions, on-road vehicle emissions, and other anthropogenic emissions. Apportionment of the overall sensitivity to VOC emissions is shown in the pie chart insets. Figures
similar to Fig. 3 but for the other receptor areas investigated here can be found in the supplementary information (SI, Fig. S17 through S19). For a more detailed apportionment of the emissions sensitivity of each model response, see Table S6 in the SI.

     In both summer and winter, the emissions sensitivity of the air basin-wide response is dominated by anthropogenic VOC emissions. The contribution from biogenic emissions is about a third of that due to anthropogenic emissions in summer; the biogenic contribution is negligible in winter. The air basin-wide response ($R$) decreases by a factor of 1.4 between summer
and winter, while its sensitivity to emissions ($\partial R/\partial E$) almost doubles. In other words, even though the population-weighted formaldehyde mixing ratio is larger in summer, scaling anthropogenic and/or biogenic emissions uniformly year-round would result in a larger mixing ratio change in winter. In practice, the effects of scaling emissions up or down should be limited to a range or perturbations where the relationship between emissions and formaldehyde mixing ratios is nearly linear. Although the boundaries of this linear range are not fully characterized, sensitivity coefficients reported here should not be applied to the
case where all emissions are removed from the simulations (i.e., a 100 % perturbation). If primary emissions of formaldehyde and precursors are removed, formaldehyde mixing ratios should approach background values of a few tens of ppb in both summer and winter. Changes in model responses would in this case be larger in summer than in winter, which contradicts our findings. The use of first-order sensitivities to characterize the behavior of a non-linear system is nevertheless applicable to identifying emissions that should be prioritized in control strategies that aim to reduce ambient formaldehyde. To summarize,
controlling anthropogenic emissions is less effective, to a first-order, at reducing mixing ratios when mixing ratios are higher (i.e. in summer). A similar seasonal trend in the emissions sensitivity is observed for the other receptor areas except for Pittsburg/Antioch, where overall emissions sensitivity varies little by season. The increase in emissions sensitivity between summer and winter is driven by the 3-fold increase in the sensitivity to formaldehyde emissions. In contrast, sensitivity to emissions of most other species is smaller in winter than in summer. In fact, almost 90 % of the sensitivity in December is
attributed to formaldehyde emissions, indicating that secondary formation of formaldehyde is negligible in winter. Secondary

formation is very significant in summer, when the influence of emissions of organic precursors is comparable to the influence of direct formaldehyde emissions. The influence of organic precursors is dominated by alkenes, of which isoprene is the largest contributor. The individual contributions of non-alkene species are all smaller than 5 %. OLE2 is the only organic precursor other than formaldehyde that consistently yields higher sensitivities in winter than in summer, meaning that controlling these reactive alkenes has a greater impact on formaldehyde during winter.

The relative contribution of direct formaldehyde emissions to the VOC emissions sensitivity in summer varies significantly across the receptor areas investigated here (Fig. 4): 46 % for the air basin as a whole, 60 % for the East Bay, 46 % for San Jose, and 24 % for Pittsburg/Antioch, located further inland/downwind. Results presented below in Section 3.5 indicate that the zones of emissions influence are larger and extend farther upwind from the receptor area for precursors compared to formaldehyde itself. As a consequence, the relative importance of precursor emissions versus directly-emitted formaldehyde increases for receptor areas located further downwind of emission-rich areas. This trend is attributed to the increased opportunity for secondary formation associated with the longer transport time required for distant emissions to reach receptors. For example, in July, the Pittsburg/Antioch receptor is relatively far ($\sim$ 50 km) downwind of many high-emission areas, including San Francisco and the East Bay. The influence of direct formaldehyde emissions on air quality at this receptor is small compared to emissions of other species. In contrast, the East Bay is $\sim$ 10 km downwind of the San Francisco peninsula, and the relative influence of direct formaldehyde emissions is considerably higher for this receptor. The San Jose and air basin-wide responses represent intermediate cases. This finding applies to both anthropogenic and biogenic emissions. In particular, the relative contribution of biogenic emissions to overall VOC emission sensitivity is largest for Pittsburg/Antioch (39 %) and smallest for the East Bay (12 %). On-road vehicle emissions of VOC account for roughly 40 % of the overall sensitivity to anthropogenic VOC emissions for all model responses investigated here (Table S6). The sensitivity of formaldehyde responses to $NO_x$ emissions is negative in both seasons for all receptor areas, with a magnitude that ranges from 10 to 28 % of the corresponding overall sensitivity to emissions. The negative sign of this sensitivity means that decreasing $NO_x$ emissions would increase formaldehyde concentrations and vice-versa. Causes of this negative sign are discussed in Section 3.4.

### 3.3   Drivers of seasonal changes in sensitivity

Seasonal changes in the emissions sensitivities discussed above cannot be explained by changes in emissions alone: emissions of formaldehyde and its precursors vary little between summer and winter in this study. A notable exception is biogenic emissions of isoprene and terpenes, which are about eight and two times higher in July than in December, respectively. Terpenes contribute little to the overall sensitivity to emissions: less than 6 % in both seasons for all model responses. Isoprene is responsible for 11 to 37 % of the overall emissions sensitivity for the July responses, but has little influence in December.

The following two key mechanisms are possible contributors to the seasonal change of sensitivity to formaldehyde emissions. First, the mixing height in the modeling domain is generally lower in winter than in summer. Formaldehyde emissions are therefore mixed within a smaller vertical column in winter, resulting in higher near-surface mixing ratios. This phenomenon, in turn, makes surface-level model responses more sensitive to formaldehyde emissions. A previous study by Bastien et al. (2015) suggests that the mixing height has a great influence on the emissions sensitivity of surface mixing ratios for primary pollutants;

it is likely that it would have a comparable influence on the sensitivity of formaldehyde mixing ratios to formaldehyde emissions. The second mechanism is seasonal variations of formaldehyde lifetime in the atmosphere. Table 2 shows month-long averages of rates of formaldehyde removal, separately for the July and December simulations, and separately for the following removal processes: photolysis, chemical reactions other than photolysis, and dry deposition. Removal by wet deposition is neglected as precipitation events are rare in the simulations. Photochemical removal decreases by one to two orders of magnitude between July and December, and removal by dry deposition decreases by a factor of 2.5 between these two periods. Dry deposition fluxes are estimated as the product of surface-layer concentrations and deposition velocities. While deposition fluxes depend on atmospheric concentrations of relevant pollutants, there are also important seasonal effects on deposition velocities. Here, the decrease in dry deposition between seasons results from the combined effects of reduced formaldehyde concentrations (Table 1) and, in some areas, reduced vertical transport and mixing within the surface layer in winter compared to summer (Figure S20).

Reduced losses by dry deposition and by photochemical reactions both contribute to a longer atmospheric lifetime of formaldehyde in winter. Table 2 shows total production and loss of formaldehyde in the modeling domain. The difference between these values is approximately equal to net transport out of the modeling domain. In winter, a larger fraction of the formaldehyde produced is transported out of the modeling domain rather than being lost due to chemical reactions or surface deposition. This trend is consistent with formaldehyde having a longer atmospheric lifetime in winter compared to summer. This extended lifetime may in turn increase the sensitivity of model responses to directly emitted formaldehyde. The influence on sensitivities of photochemical removal of formaldehyde is investigated by running model simulations in which formaldehyde is treated as a tracer species: chemistry is turned off and there is neither secondary formation nor photochemical removal of formaldehyde. In these simulations, removal of formaldehyde is due only to dry deposition and transport out of the modeling domain. The atmospheric lifetime of formaldehyde therefore increases. When the residence time of a particular molecule increases, on average these molecules stay in the modeling domain for longer periods of time and their concentrations increase. In other words, sensitivities of model responses to formaldehyde emissions are expected to be larger when chemistry is turned off. The goals of these additional simulations are to (i) characterize the influence of photochemical removal on sensitivities of model responses to formaldehyde emissions, and (ii) to investigate whether the large (∼200 %) increase of these sensitivities from July to December can be explained by the reduction of photochemical removal between these two seasons. With chemistry turned off in the model, formaldehyde emissions sensitivities increased by 9 to 28 % in July, but by only a few percent in December (Table 3). Although the relative increase in sensitivity is larger in July for all receptor areas, the actual magnitude of this change is similar in July and December for two of these receptors (air basin and San Jose), because sensitivities to formaldehyde emissions are larger in December. Overall, seasonal variations in photochemical removal rates of formaldehyde do not explain the large (∼200 %) increase in formaldehyde emissions sensitivity of model responses from July to December. These results suggest that reduced transport and mixing are the main drivers of the enhanced influence on air quality of directly-emitted formaldehyde in December compared to July.

## 3.4 Sensitivities to reaction rate coefficients

Comparing sensitivities to reaction rate coefficients gives insights into which species and chemical reactions contribute the most to photochemical production and removal of formaldehyde. Rate sensitivities for each model response and for each of the 211 chemical reactions represented in the SAPRC99 chemical mechanism were calculated and ranked by magnitude. Table 4 shows the 16 reactions with the largest sensitivities for the air basin-wide response, separately for July and December. A negative rate sensitivity indicates that increasing the reaction rate coefficient would decrease the formaldehyde response, and vice-versa. It should be noted that ordering reactions by rate sensitivity versus ordering them by reaction rate yields different rankings. Rate sensitivities account for both the rate and the formaldehyde yield of the reaction. In other words, they account for how often the reaction occurs and how many molecules of formaldehyde are eventually formed or destroyed every time the reaction occurs. The rate coefficient sensitivities presented in Table 4 do not indicate, however, which reactions produce the most formaldehyde. Rather, they indicate how much formaldehyde concentrations would be affected if rate coefficients were to change. Only semi-normalized sensitivities are considered here. Non-normalized sensitivities cannot be ranked in the same manner because they have different dimensions depending on whether the reaction is uni-, bi-, or termolecular.

We first discuss the summertime results. In this season, attack by the OH radical (R3) is the highest-ranking reaction among the five photochemical pathways for formaldehyde removal represented in the model. While the average formaldehyde photolysis rate is only 45 % larger for the channel that yields stable products (R2) than for the channel that yields radicals (R1), the rate sensitivity corresponding to R2 is about twice as large in magnitude as for R1. The reason for this difference is that the $HO_2$ radicals generated as products of R1 can themselves participate in secondary formation of formaldehyde, thus partially offsetting the initial loss of formaldehyde. The net effect of the reaction is nevertheless a loss of formaldehyde, as indicated by the negative sign of the corresponding sensitivity. The OH radical can remove formaldehyde from the atmosphere, but it can also oxidize other organic species to eventually form more formaldehyde. The latter effect prevails over the former, as indicated by the positive rate sensitivity of the OH-forming reaction involving $H_2O$ and $O(^1D)$.

The highest-ranking reaction that yields a positive sensitivity is the reaction of methane with the OH radical. This reaction leads to the formation of methyl peroxy radical, which can itself lead to formaldehyde. Other high-ranking pathways to secondary formaldehyde are attack of methanol by the OH radical, and the photolyses of methacrolein, glyoxal, and methyl vinyl ketone. In Section 3.2, alkenes – more specifically ISOPRENE, ETHENE, OLE1, and OLE2 – were identified as the main precursors to secondary formaldehyde in July. Yet, none of the reactions that directly involve any of these species appear in Table 4 for this season (the highest-ranking one among these reactions – reaction of ethene with OH – ranks 18[th]). These two seemingly contradictory observations suggest that these alkenes are reactive enough that the factor that limits formation of formaldehyde is not the rate at which the reactions occur (low sensitivity to reaction rates) but rather the availability of the alkenes as reactants (high sensitivity to emissions). For species such as glyoxal and methacrolein, emissions are small but secondary formation is large. These species do contribute significantly to secondary formation of formaldehyde, yet the corresponding emissions sensitivities are small because these species are formed chemically within rather than being directly emitted into the atmosphere; they can also be transported from upwind regions.

We note the opposite effects on formaldehyde of two competing fates for peroxyacetyl radicals ($CH_3C(O)OO^\bullet$). The corresponding sensitivities are of comparable magnitudes but opposite signs. Reaction of these radicals with NO forms $CO_2$ and methyl peroxy radicals ($CH_3OO^\bullet$). The latter are readily converted to formaldehyde. The corresponding rate sensitivity is, accordingly, positive. In contrast, reaction of peroxyacetyl radicals with $NO_2$ forms peroxyacetyl nitrate (PAN). PAN is rela-

5 tively stable in winter when temperature is low. PAN can dissociate to release $CH_3C(O)OO^\bullet$ and $NO_2$ back to the atmosphere. As long as peroxyacetyl radicals and $NO_2$ are bound together as PAN, they cannot react with other species and contribute to formaldehyde production. The PAN-forming reaction therefore inhibits production of secondary formaldehyde, and the corresponding rate sensitivity is negative. More generally, secondary formation of formaldehyde is enhanced by photolysis of $NO_2$, whereas it is inhibited by NO to $NO_2$ conversions associated with the reaction of NO with ozone. NO contributes to the

10 formation of formaldehyde by converting a number of different peroxy radicals into alkoxy radicals which can be further converted to formaldehyde (Seinfeld and Pandis, 2016). $NO_2$ inhibits formation of formaldehyde by reacting with OH radicals to form nitric acid and with peroxyacetyl radicals to form PAN. The removal of OH radicals by the $OH + NO_2$ chain-terminating reaction is likely responsible for the negative sign of the sensitivities of formaldehyde to $NO_x$ emissions (Section 3.2).

As discussed previously, secondary formation of formaldehyde is far less significant in December than in July. This trend is

15 also reflected in the high-ranking rate sensitivities, most of which are smaller in magnitude in December.

### 3.5 Spatial distributions of sensitivities

The apportionment analysis presented in Section 3.2 identifies the chemical species whose emissions have a large influence on formaldehyde mixing ratios within the study domain. To mitigate air pollution problems effectively, the influence of emissions as a function of location and time must be understood. The influence of a source depends not only on its strength, but also

on parameters such as location relative to receptor, meteorological conditions, and the chemical state of the atmosphere. This section investigates how the spatial distribution of emission sources affects their influence on air pollution within different receptor areas. We also consider whether the magnitude of an emission source correlates with the influence of that source.

Figure 5 shows the relationship between the magnitude of emissions of selected species and their influence on the air basin-wide (left column) and East Bay (right column) responses, separately for July and December. Figure 5 also shows, for each

25 model response, each model species, and each simulation period, the corresponding linear regression line and coefficient of determination, $r^2$. The formaldehyde emissions sensitivity of the air basin-wide model response correlates only moderately well with the magnitude of these emissions, both in July ($r^2 = 0.46$) and December ($r^2 = 0.59$). The largest formaldehyde emission source in the modeling domain is located near Fairfield. As this source is comparatively far from and often downwind of the most populated areas of the air basin, it contributes little to the air basin-wide response, and thus drives down the corresponding

correlation between emissions and sensitivity. Removing this point from the analysis yields better correlations ($r^2 = 0.70$ and 0.76 in July and December, respectively). There is no correlation between the magnitude of formaldehyde emission sources and the corresponding sensitivities for any of the more local model responses investigated in this study: Pittsburg/Antioch, San Jose, and the East Bay. For example, $r^2 = 0.18$ and 0.12 in July and December, respectively, for the East Bay (Fig. 5b). As

discussed further below, the location of emission sources relative to the receptor becomes crucial in determining their effects on air quality within localized receptor areas.

The sensitivity of model responses to anthropogenic emissions of organic precursors correlates more strongly with the magnitude of these emissions than was the case for formaldehyde emissions. This finding holds for the air basin-wide response and for all the local responses under investigation, with stronger correlations for the air basin-wide response than for the local responses. Examples are shown on Fig. 5c (air basin-wide response) and 5d (East Bay response) for ethene. The strength of the correlations between the $NO_x$ emissions sensitivity of formaldehyde model responses and the magnitude of these emissions (Fig. 5e and 5f) is similar to that observed for anthropogenic emissions of organic precursors. The correlation between sensitivity to biogenic emissions and the magnitude of these emissions is weak in most cases, as illustrated by Figures 5g and 5h for isoprene. There are large biogenic emission source regions which do not contribute significantly to local and regional formaldehyde exposure, while other smaller sources, located upwind of densely populated areas, have a much greater influence.

The spatial distributions of anthropogenic emissions are similar for most model species, and the strength of the correlations discussed here gives an indication of how strongly the influence of emission sources depends on their location relative to the receptor area. A weak correlation suggests a high dependence on location. Results discussed above suggest that the location of emission sources has a greater influence for directly-emitted formaldehyde than for anthropogenic precursors to secondary formation. This conclusion is further supported by the spatial distributions of sensitivity to emissions of formaldehyde and other precursors, as discussed below. Non-normalized sensitivity to emissions at location $x$ measures the change in model response resulting from adding new emissions at that location, where the amount of pollutant thus added is independent of the magnitude of existing emissions. In other words, it measures how much a certain location can affect a specified model response. A spatially invariant non-normalized sensitivity means that the influence that emissions have on air quality does not depend on their location, only on their magnitude. If non-normalized sensitivity is not uniform, then the magnitude of its gradient indicates how much the location of emissions matters in determining their impact on air quality.

Figures 6a and 6b show the non-normalized sensitivity for July of the San Jose response to formaldehyde and ethene emissions, respectively. The sensitivity to formaldehyde emissions exhibits strong spatial gradients, indicating that the influence of direct formaldehyde emissions strongly depends on location. In contrast, the sensitivity to ethene emissions exhibits weaker spatial gradients over a larger region, especially along the prevailing direction of surface winds (roughly northwest to southeast in the case of San Jose). The influence of ethene emissions is not as dependent on the location of the emissions compared to formaldehyde. Similar features are observed in the spatial distributions of non-normalized sensitivities for the other summer-season responses.

The contribution of an existing emission source to a model response is measured by the corresponding semi-normalized sensitivity, which is calculated as the product of the potential influence of the location of this source and its magnitude. Mathematically:

$$S_E^\times = S_E^+ \times E \tag{10}$$

where the three terms are, respectively: semi-normalized sensitivity to emissions, non-normalized sensitivity to emissions, and magnitude of emissions. Figures 6c and 6d show the semi-normalized sensitivity for July of the San Jose response to formaldehyde and ethene emissions, respectively. The influence of formaldehyde emissions is dominated by local sources while the influence of ethene emissions is more affected by sources located further upwind of the receptor area.

Similar observations can be made for the other local responses investigated in this study. For example, Fig. 7a and 7b show the spatial distributions of semi-normalized sensitivity of the Pittsburg/Antioch response in July to emissions of formaldehyde and VOC precursors, respectively. Precursor emissions originating from San Francisco have a strong influence on formaldehyde in Pittsburg/Antioch even though these emissions are located more than 50 km away from the receptor. Precursor emissions originating from the East Bay and from within the receptor itself, as well as from regions along this path also have a significant influence on this response. In contrast, formaldehyde emissions originating from San Francisco and the East Bay do not contribute as much as local formaldehyde emissions to the sensitivity of the Pittsburg/Antioch response. The same trend is observed in December (Fig. 7e and 7f)

Compared to summer, the relative contribution of sources located within or near the receptor area is larger in winter for both formaldehyde and ethene emissions. This same seasonal pattern was also noted for benzene and diesel black carbon (Bastien et al., 2015), and was attributed to differences in meteorology between the two seasons. In July, near-surface winds are predominantly from the northwest or west. In December, near-surface winds are weaker and do not have a clear prevailing direction. The more quiescent meteorological conditions associated with the winter season increase the relative importance of nearby sources. Relevant upwind areas of influence extend west of the receptor areas in July and to a lesser extent east of the receptor areas in December. The relative contribution of local versus upwind formaldehyde emissions increases significantly for the Pittsburg/Antioch receptor between July and December (44 to 77 %), decreases for San Jose (73 to 62 %), and is 74 % in both seasons for the East Bay. As an additional example, Figures 7c and 7g show the formaldehyde emissions that influence formaldehyde in the East Bay in July and December, respectively. Emissions originating from San Francisco have more influence on formaldehyde in the East Bay in July compared to December, and East Bay emissions become more influential in December. Figure 7d shows the semi-normalized sensitivity of the air basin-wide July response to $NO_x$ emissions, which is negative in and around the most densely populated areas but is positive for highway emissions in the North and for ship emissions over the Pacific ocean. These positive $NO_x$ sensitivities are, however, small in magnitude (see also Fig. 5e).

## 4   Conclusions

The adjoint of a chemical transport model is used to identify emission sources and chemical reactions that have the most influence on formaldehyde, separately for summer- and winter-season conditions. Air quality metrics investigated in this work are population-weighted mixing ratios in the San Francisco Bay Area as a whole and within three of its urbanized sub-areas. The results of the sensitivity analysis are used to quantify the relative importance of primary versus secondary formation, anthropogenic versus biogenic sources, and of different emission sectors on formaldehyde pollution. Secondary formation has little influence on formaldehyde in winter: 85–90 % of the sensitivity of formaldehyde to emissions is attributed to directly-emitted

formaldehyde, and 62–77 % of this contribution is attributed to local sources. In summer, the relative contribution of direct formaldehyde emissions is more variable and ranges between 27 and 75 %. The remainder of the formaldehyde sensitivity is attributed to emissions of precursors, mainly isoprene, ethene, and other alkenes. Areas of influence extend farther upwind for precursors than for direct emissions of formaldehyde. The relative influence of secondary formation versus direct emissions

on formaldehyde is largest for receptors located farther away from heavily urbanized and emission-rich areas. The relative importance of biogenic emissions is also largest for these receptors. This pattern is attributed to the increased opportunity for secondary formation associated with the longer transport time required for polluted air masses to reach distant receptors. Increasing $NO_x$ emissions decreases formaldehyde pollution at receptors (and, conversely, decreasing $NO_x$ emissions increases formaldehyde pollution), but the magnitude of this effect is relatively small. To summarize, winter-season formaldehyde is

greatly influenced by direct local emissions of formaldehyde. In contrast, summer-season formaldehyde is influenced by both direct local emissions of formaldehyde and regional emissions of other organic precursors. The magnitude of anthropogenic emissions of species other than formaldehyde correlates reasonably well with their influence on regional formaldehyde exposure. They do not, however, correlate with their influence on pollution within specific urban sub-areas.

    This study uses first-order sensitivity coefficients to characterize how formaldehyde at different receptors responds to

changes in emissions and reaction rate coefficients. The use of first-order sensitivity coefficients to estimate changes in responses of non-linear systems (Eq. 8 and 9) while ignoring higher-order terms of the Taylor series (Eq. 5) is subject to restrictions. The changes in parameters must be relatively small so that the system response remains approximately linear. The first-order sensitivities presented here should therefore not be used to estimate the effects on ambient formaldehyde of very large emission changes such as 100 % reductions. Perturbing NO emissions was found to yield particularly non-linear behavior.

Additionally, first-order sensitivities should be recalculated if emissions or other model parameters were to change significantly from the baseline values used here. This study nevertheless provides a basis to identifying the emissions and chemical reactions that have the most influence on ambient formaldehyde.

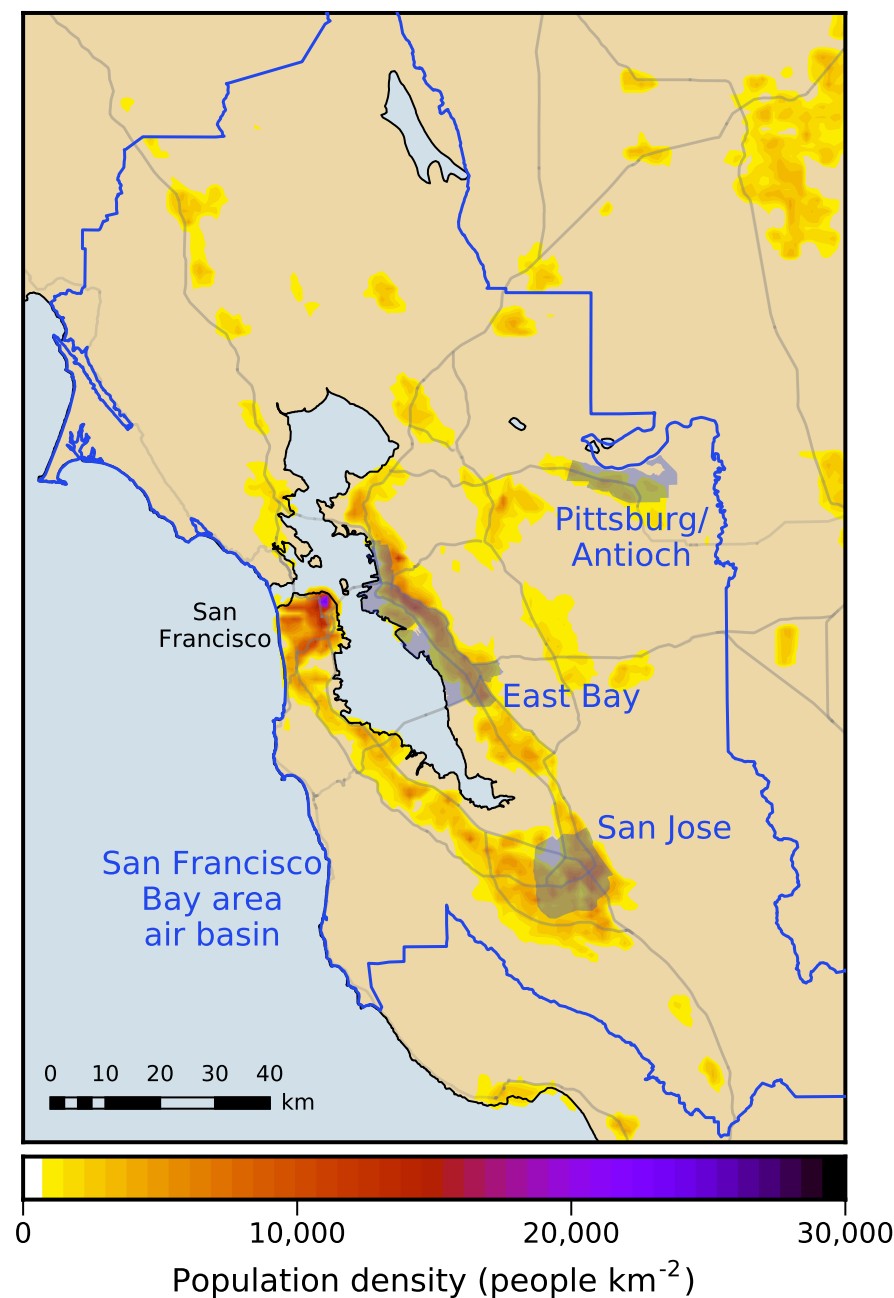

**Figure 1.** Modeling domain and receptor areas investigated in this work. The color scale shows population density as of the 2010 U.S. Census. Gray lines indicate major roadways.

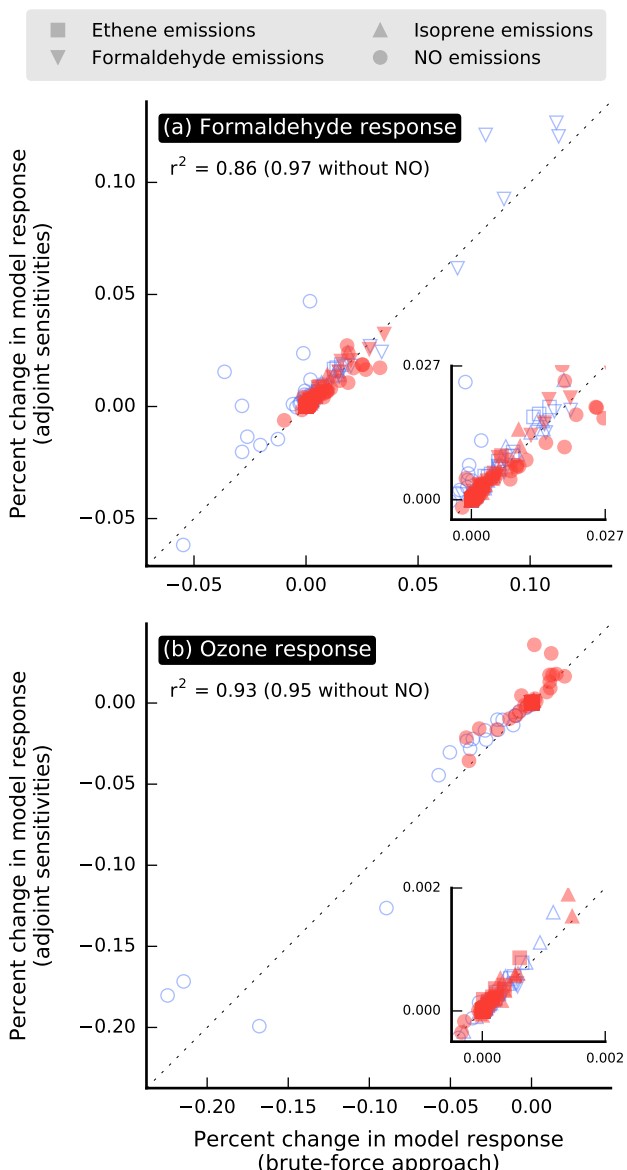

**Figure 2.** Relative change in model response calculated from first-order adjoint sensitivities versus using a brute-force approach. The model responses used for the evaluation are domain-wide population-weighted mixing ratios of (a) formaldehyde and (b) ozone. Simulation periods used for this evaluation are one-day episodes in July (red filled symbols) and December (blue unfilled symbols). Each point corresponds to the effect of an additive 24-hour perturbation in emissions applied to a single grid cell. Insets zoom in on a section of the corresponding plots.

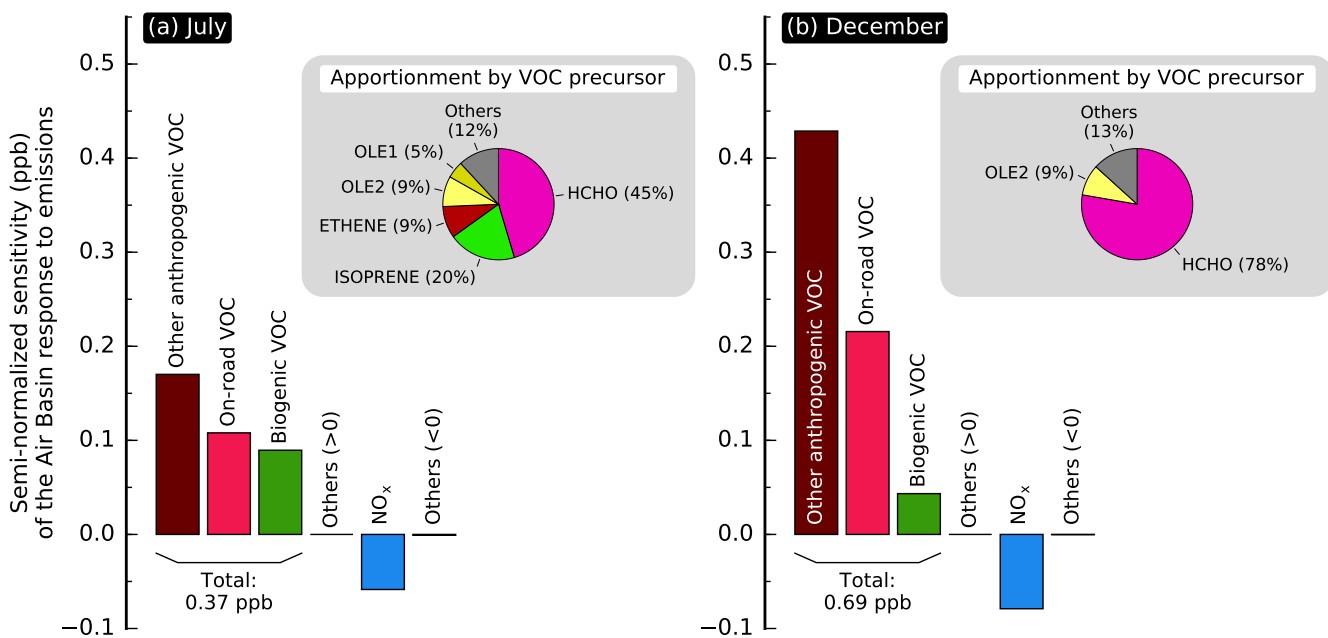

**Figure 3.** Semi-normalized sensitivity of the air basin-wide population-weighted formaldehyde mixing ratio to emissions, in (a) July and (b) December. An apportionment by precursor species of the sensitivity to overall VOC emissions is shown in the pie chart insets.

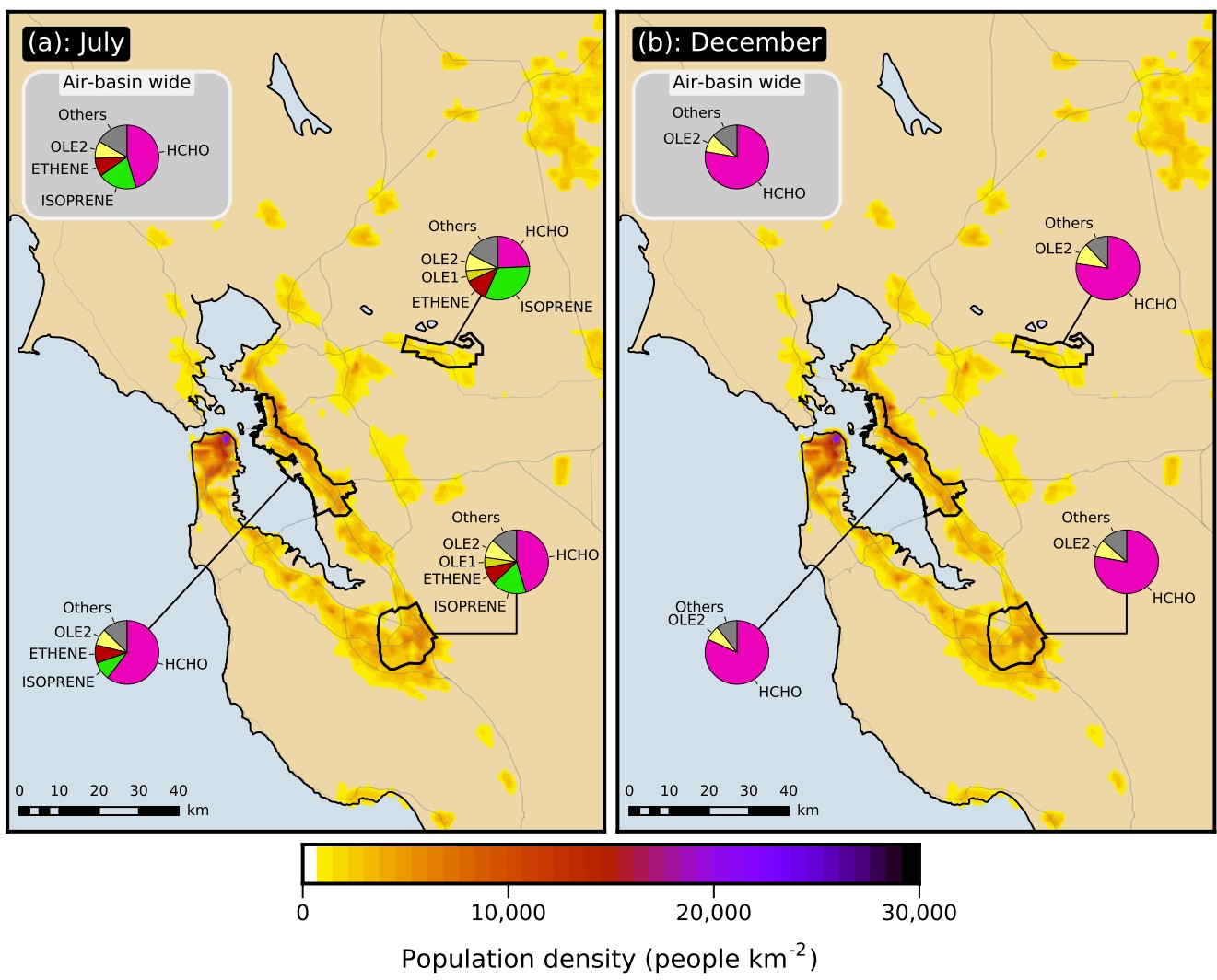

**Figure 4.** Apportionment by model species (pie charts) of the VOC emissions semi-normalized sensitivity of the formaldehyde response in the San Francisco Bay Area air basin as a whole and within three of its urbanized sub-areas in (a) July and (b) December.

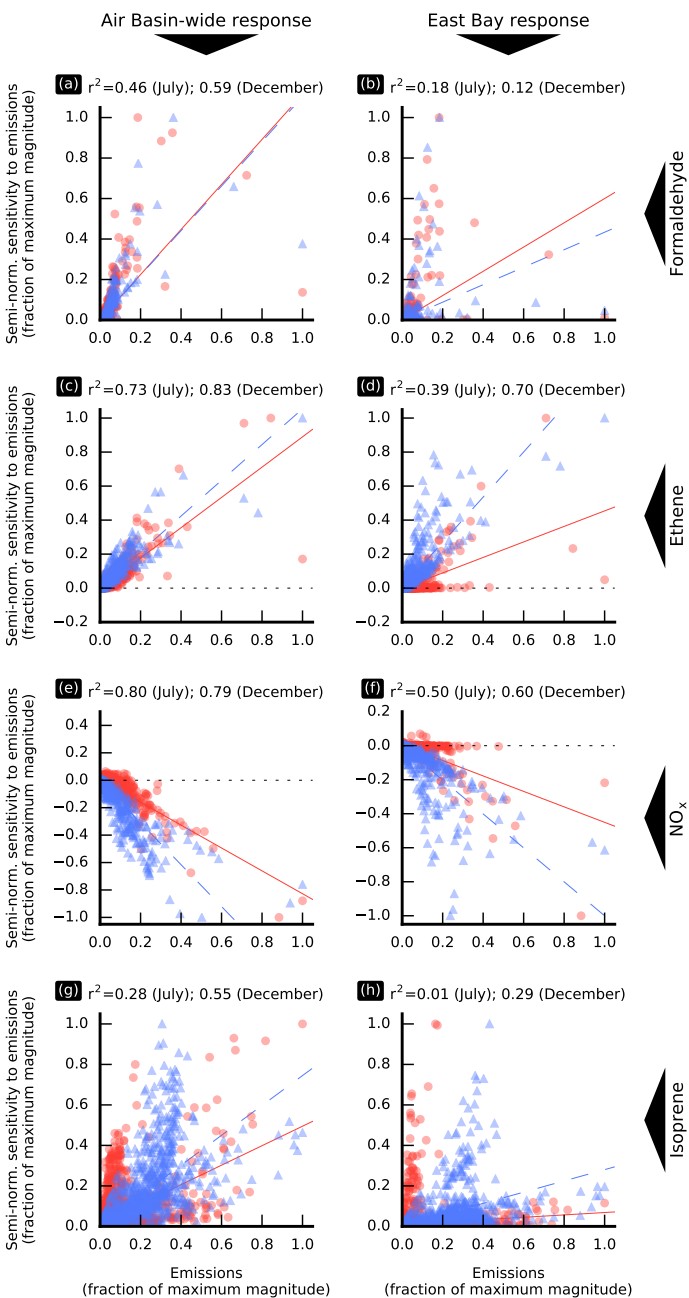

**Figure 5.** Semi-normalized sensitivity of model responses to individual emission sources versus the magnitude of these sources. Values are shown separately for the July (red circles) and December (blue triangles) simulation periods, along with the corresponding linear regression affine lines (solid red line for July and dashed blue line for December). The corresponding coefficients of determination, $r^2$, are indicated on the top of each subplot. In this figure, an "individual emission source" corresponds to emissions of a model species in a single model grid cell. Values are averaged over the simulation period.

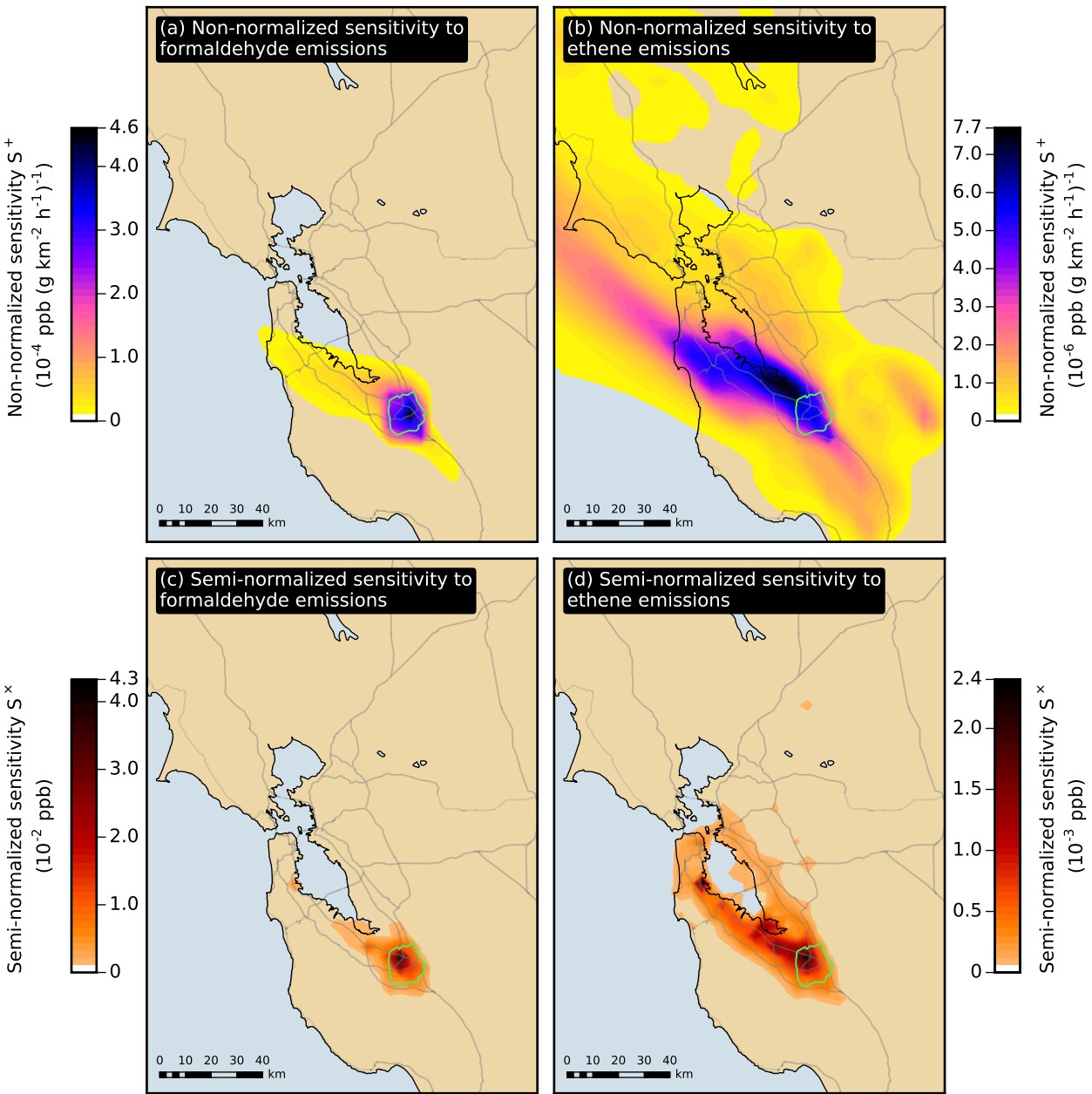

**Figure 6.** Non-normalized and semi-normalized sensitivity of the San Jose formaldehyde response in July to emissions of formaldehyde and ethene. At each location, the sensitivity is summed over the duration of the simulation period. The San Jose receptor area is outlined in green.

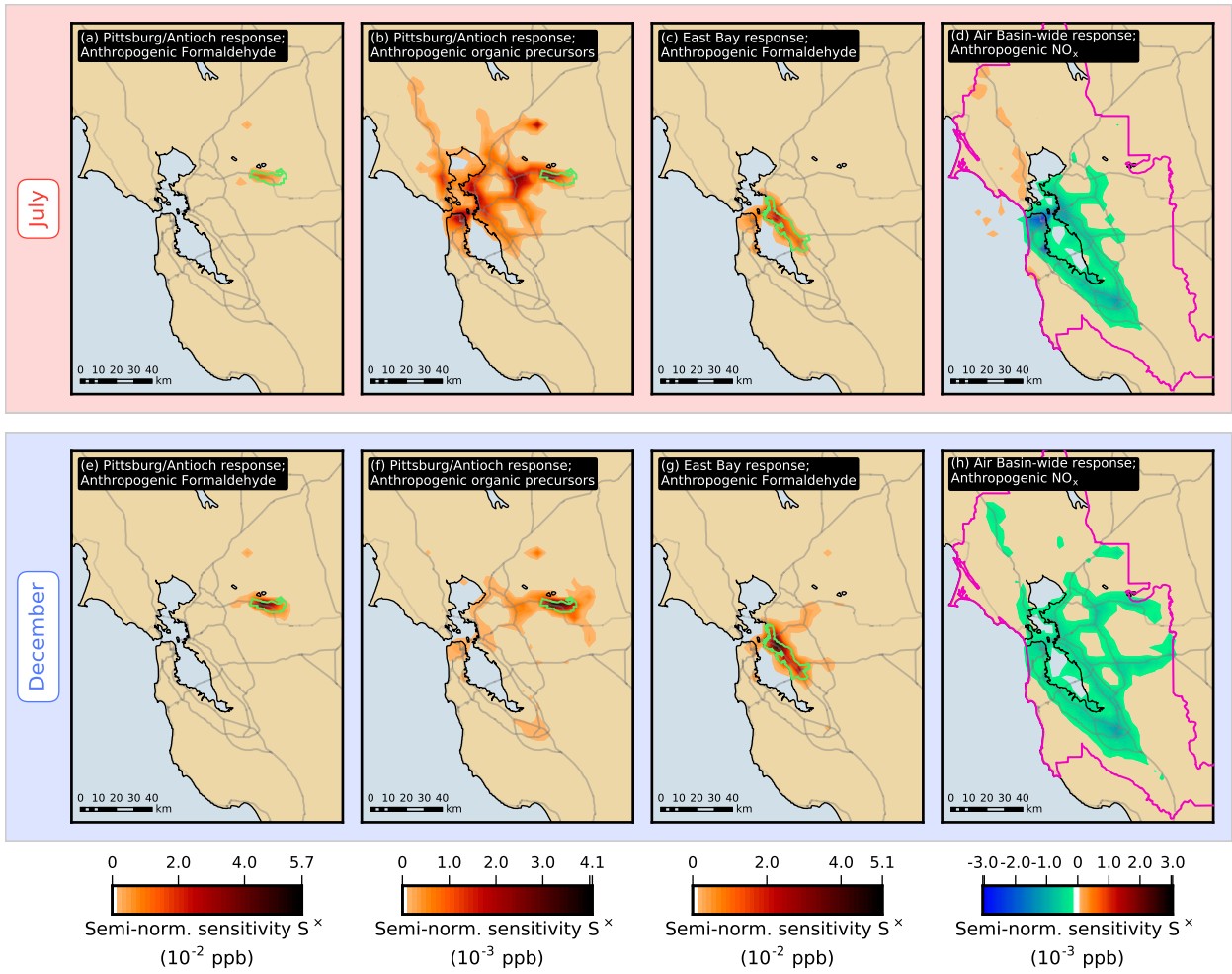

**Figure 7.** Semi-normalized sensitivity $S^{\times}$ of the formaldehyde model response at different receptors to emissions of various species or groups of species. In each grid cell, sensitivity values are summed over the duration of the simulation period. The boundary of the receptor area under consideration in each case is indicated by a green or pink line.

**Table 1.** Model responses in ppb. Model responses are population-weighted monthly average formaldehyde mixing ratios for the San Francisco Bay Area (air basin) and selected sub-areas.

|          | Air Basin | Pittsburg/Antioch | San Jose | East Bay |
|----------|-----------|-------------------|----------|----------|
| July     | 1.3       | 1.5               | 1.5      | 1.4      |
| December | 0.92      | 0.59              | 1.3      | 0.94     |

**Table 2.** Monthly average production and loss rates of formaldehyde by different mechanisms. Values are summed over the modeling domain.

|                                   | Emissions $(\mathrm{mol\,s^{-1}})$ | Chemical production $(\mathrm{mol\,s^{-1}})$ | Loss by photolysis $(\mathrm{mol\,s^{-1}})$ | Loss by other reactions $(\mathrm{mol\,s^{-1}})$ | Loss by dry deposition $(\mathrm{mol\,s^{-1}})$ | Total production $(\mathrm{mol\,s^{-1}})$ | Total loss $(\mathrm{mol\,s^{-1}})$ |
|-----------------------------------|-----------|-----------|-----------|-----------|-----------|-----------|-----------|
| **Surface-layer only:**           |           |           |           |           |           |           |           |
| July                              | 2.0       | 5.4       | 1.2       | 1.9       | 6.8       |           |           |
| December                          | 2.0       | 0.66      | 0.091     | 0.063     | 2.7       |           |           |
| **Planetary boundary layer only:** |          |           |           |           |           |           |           |
| July                              | 2.0       | 160       | 45        | 68        | 6.8       |           |           |
| December                          | 2.0       | 5.0       | 0.89      | 0.57      | 2.7       |           |           |
| **All layers:**                   |           |           |           |           |           |           |           |
| July                              | 2.0       | 370       | 140       | 160       | 6.8       | 370       | 310       |
| December                          | 2.0       | 49        | 16        | 11        | 2.7       | 51        | 30        |

**Table 3.** Increase (ppb) in the formaldehyde emissions semi-normalized sensitivity of model responses resulting from turning off chemistry in the model. The numbers in parentheses indicate the magnitude of the increase relative to the original value.

|          | Air Basin     | Pittsburg/Antioch | San Jose      | East Bay       |
|----------|---------------|-------------------|---------------|----------------|
| July     | 0.020 (12%)   | 0.024 (28%)       | 0.031 (12%)   | 0.026 (9.3%)   |
| December | 0.018 (3.3%)  | 0.006 (2.2%)      | 0.038 (4.2%)  | 0.013 (2.4%)   |

**Table 4.** Reaction rates (averaged over the modeling domain and simulation period) and semi-normalized sensitivities $S_k^{\times}$ (summed over the modeling domain and simulation period) of the air basin-wide response to reaction rate coefficients for the 16 reactions for which $S_k^{\times}$ has the largest magnitude. Reactions are ranked separately for each season.

| Rank | Reaction[a] | Average reaction rate (molecule cm$^{-3}$ s$^{-1}$) | Actual sensitivity to reaction rate coefficient (ppb) |
|---|---|---|---|
| **July** | | | |
| 1 | HCHO + OH $\rightarrow$ HO$_2$ + CO + H$_2$O | $7.5\times10^{+5}$ | $-0.42$ |
| 2 | CH$_4$ + OH $\rightarrow$ CH$_3$OO$^{\bullet}$ + H$_2$O | $6.3\times10^{+5}$ | $+0.30$ |
| 3 | HCHO + h$\nu$ $\rightarrow$ CO + H$_2$ | $4.5\times10^{+5}$ | $-0.25$ |
| 4 | Methacrolein + h$\nu$ $\rightarrow$ 0.67HCHO + 0.34HO$_2$ + ... | $2.9\times10^{+5}$ | $+0.23$ |
| 5 | NO$_2$ + h$\nu$ $\rightarrow$ NO + O($^3$P) | $6.8\times10^{+7}$ | $+0.23$ |
| 6 | O$_3$ + NO $\rightarrow$ NO$_2$ + O2 | $6.1\times10^{+7}$ | $-0.19$ |
| 7 | Methanol + OH $\rightarrow$ HCHO + HO$_2$ + H$_2$O | $4.0\times10^{+5}$ | $+0.19$ |
| 8 | O$_3$ + h$\nu$ $\rightarrow$ O($^1$D) + O2 | $1.2\times10^{+7}$ | $+0.17$ |
| 9 | O($^1$D) + M $\rightarrow$ O($^3$P) + M | $1.2\times10^{+7}$ | $-0.16$ |
| 10 | OH + NO$_2$ $\rightarrow$ HNO$_3$ | $5.0\times10^{+5}$ | $-0.14$ |
| 11 | Glyoxal + h$\nu$ $\rightarrow$ HCHO + CO | $2.3\times10^{+5}$ | $+0.13$ |
| 12 | HCHO + h$\nu$ $\rightarrow$ 2HO$_2$ + CO | $3.1\times10^{+5}$ | $-0.12$ |
| 13 | O($^1$D) + H$_2$O $\rightarrow$ 2OH | $6.7\times10^{+5}$ | $+0.09$ |
| 14 | Methylvinylketone + h$\nu$ $\rightarrow$ 0.3CH$_3$OO$^{\bullet}$ + 0.7CO ... | $5.3\times10^{+5}$ | $+0.08$ |
| 15 | CH$_3$C(O)OO$^{\bullet}$ + NO $\rightarrow$ CH$_3$OO$^{\bullet}$ + NO$_2$ + CO$_2$ | $4.4\times10^{+5}$ | $+0.08$ |
| 16 | CH$_3$C(O)OO$^{\bullet}$ + NO$_2$ $\rightarrow$ PAN | $9.0\times10^{+5}$ | $-0.07$ |
| **December** | | | |
| 1 | Glyoxal + h$\nu$ $\rightarrow$ HCHO + CO | $8.4\times10^{+4}$ | $+0.14$ |
| 2 | NO$_2$ + h$\nu$ $\rightarrow$ NO + O($^3$P) | $4.5\times10^{+7}$ | $+0.14$ |
| 3 | O$_3$ + h$\nu$ $\rightarrow$ O($^1$D) + O2 | $1.6\times10^{+6}$ | $+0.14$ |
| 4 | O($^1$D) + M $\rightarrow$ O($^3$P) + M | $1.4\times10^{+6}$ | $-0.12$ |
| 5 | O$_3$ + NO $\rightarrow$ NO$_2$ + O2 | $4.0\times10^{+7}$ | $-0.11$ |
| 6 | CH$_4$ + OH $\rightarrow$ CH$_3$OO$^{\bullet}$ + H$_2$O | $7.1\times10^{+4}$ | $+0.09$ |
| 7 | OH + NO$_2$ $\rightarrow$ HNO$_3$ | $8.6\times10^{+4}$ | $-0.08$ |
| 8 | HCHO + h$\nu$ $\rightarrow$ CO + H$_2$ | $4.1\times10^{+4}$ | $-0.08$ |
| 9 | Methacrolein + h$\nu$ $\rightarrow$ 0.67HCHO + 0.34HO$_2$ + ... | $2.3\times10^{+4}$ | $+0.08$ |
| 10 | CH$_3$C(O)OO$^{\bullet}$ + NO$_2$ $\rightarrow$ PAN | $2.3\times10^{+5}$ | $-0.05$ |
| 11 | OLE2 + O$_3$ $\rightarrow$ 0.269HCHO + 0.378OH + 0.003HO$_2$ ... | $3.1\times10^{+4}$ | $+0.05$ |
| 12 | HCHO + OH $\rightarrow$ HO$_2$ + CO + H$_2$O | $3.0\times10^{+4}$ | $-0.05$ |
| 13 | CH$_3$C(O)OO$^{\bullet}$ + NO $\rightarrow$ CH$_3$OO$^{\bullet}$ + NO$_2$ + CO$_2$ | $4.2\times10^{+4}$ | $+0.05$ |
| 14 | PAN $\rightarrow$ CH$_3$C(O)OO$^{\bullet}$ + NO$_2$ | $2.1\times10^{+5}$ | $+0.04$ |
| 15 | O($^1$D) + H$_2$O $\rightarrow$ 2OH | $5.0\times10^{+4}$ | $+0.04$ |
| 16 | ETHENE + OH $\rightarrow$ 1.61HCHO + RO2_R + 0.195CCHO | $1.1\times10^{+4}$ | $+0.03$ |

[a] O$_2$ may be omitted as a reactant. PAN is peroxyacetyl nitrate.

*Code and data availability.* The model source code is included in the supplementary material. Data sets used in this study will be provided upon request addressed by email to the corresponding author.

*Author contributions.* LAJB conducted the modeling work, data analysis, and manuscript write-up under the supervision and with the advising of RAH and NJB.

5 *Competing interests.* The authors declare no competing interests.

*Acknowledgements.* Financial support for this work was provided, in part, by the Bay Area Air Quality Management District (BAAQMD), the Carl W. Johnson Foundation, and the Director, Office of Science, Office of Basic Energy Sciences, Chemical Sciences, Geosciences, and Biosciences Division of the U.S. Department of Energy, under contract No. DE-AC02-05CH11231. We thank Philip Martien, Yiqin Jia, and Cuong Tran from BAAQMD for their support and assistance.

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
