# Peer review of "Contributions to local and regional-scale formaldehyde concentrations"

_Atmospheric Chemistry and Physics, 2018_

## Referee Comment (RC1) · Anonymous Referee #2 · 10 Jan 2019

General comments:

The manuscript is very well structured and written. The manuscript is very interesting, adding more information on the use of the adjoint technique, in this case to calculate the sensitivity of formaldehyde to emissions of NOx, formaldehyde, and VOC precursors. I think the article deserves publication. My only concern is related to the linearity approach as I explain in my comments below.

Specific comments:

1. Please identify the radicals in R1 and R3 with more conventional radical identifiers.

2. Pg 6. "These studies suggest that ozone responses vary nearly linearly with emissions of precursor species for perturbations up to about 15–20 % of the nominal value."

[Figure]

Could the authors discuss this a bit more? Does this apply also for NOx emissions?

3. Pg. 6. The authors assume that (for formaldehyde) the "responses investigated here also vary approximately linearly with precursor emissions for perturbations up to about 15–20 % ". Even considering a potential linearity in O3 response in the suggested range, the chemistry of formaldehyde is different. To support this second approach (similarity to O3), could the authors include (if possible and available) more discussion, based on other studies, chambers, etc.?

4. S 3.1. l-30 "The agreement is less satisfactory for perturbations in NO emissions". This is in line with my previous comment, on the hypothesis of the linearity of the formaldehyde response to precursor emissions. The comparison with the brute force approach indicates that this assumption of linearity is not completely valid. The conclusion section should include some comments on this uncertainty, to help the reader to interpret these results and to figure out the aspects that could be affected by this assumption of linearity.

---

## Referee Comment (RC2) · Parrish (Referee) · 12 Jan 2019

Summary:

The paper presents an interesting analysis of formaldehyde sources in the San Francisco Bay area. This research is relevant because 1) formaldehyde is an air toxic to which significant health impacts have been attributed, and 2) the modeling methods presented here may well have wider applications. The study is well conceived and executed, and the paper is well written, with the procedures and results concisely and clearly explained. I recommend that the paper be published when the following comments are addressed. It is most important to address the major comments that question if everything is done correctly (major comments 1, 3, 6, 7) and less important to

address the comments related to more discussion of formaldehyde sources and sinks.

Major issues and comments:

1) Section 2, which describes the model, the procedures for conducting the modeling, and the interpretation of the model results, is largely outside my area of expertise, so I can neither support nor critique most of that material. Section 3.1 provides the only evaluation of the model performance that is accessible to me, and I expect that this is true of most readers of the paper. The evaluation of the forward model performance is only through comparisons of monthly average diurnal cycles of mixing ratios of O3, NO and NO2 (Figures S2-S7) for the July and December simulation periods. However, the discussion is qualitative (e.g., "Agreement is good for O3 mixing ratios in July and December ....."), and in my view, not entirely accurate. Some large errors in the O3 simulation are evident, so that I am not confident that the model simulation can be accurately used for its intended purpose. I recommend a more thoughtful evaluation be conducted that can provide some quantitative information regarding how well the model actually performed. In this regard, I have the following suggestions.

The three species O3, NO and NO2 are closely coupled in the atmosphere due to rapid interconversion: $NO + O_3 \Longrightarrow NO_2 + O_2$; $NO_2 + h\nu + O_2 \Longrightarrow NO + O_3$. Hence, it makes sense to base the model-observation comparison on NOx = NO + NO2 and Ox = O3 + NO2 (which are not affected by the above interconversion) rather than on the separate species, whose concentrations are more difficult to interpret.

For O3 at any low elevation site in the Bay Area in the absence of continental (natural or anthropogenic) effects, I would expect a constant average diurnal profile at about 25 ppb in summer and 40 ppb in winter (see Parrish et al., 2009). The utility of the model is to simulate how concentrations differ from this zero-order expectation. To objectively evaluate the model performance, one should quantitatively compare how the simulated and observed Ox concentrations differ from this expectation. A qualitative examination of the figures suggests that the model may be substantially overestimating O3 (and

therefore Ox) production in the SF Bay area in summer. If there is a quantitative over-estimate in the simulated photochemical production of ozone, then the implications for the accuracy of the simulation of the photochemical production of formaldehyde must be discussed.

The observed NOx concentrations are expected to reflect anthropogenic emissions, so disagreements in a quantitative comparison of the simulated and observed NOx (or CO) concentrations may indicate errors in the emission inventory or in the simulation of vertical transport and horizontal advection. If the comparison is poor, then the reasons for the poor comparisons, and the implications for the conclusions of this paper must be discussed.

For the carbonyls, I cannot recommend an approach superior to that taken in Figures S8-S10, but the results can be more informatively summarized. Since one goal of this paper is to understand formaldehyde concentrations, any bias in the model simulations of the aldehydes should be quantitatively discussed. A qualitative examination of the figures suggests that there is no significant bias in the formaldehyde results, but that acetaldehyde simulations may be biased high; these conclusions should be put on a quantitative basis. The model underestimate of acetone mixing ratios should also be put on a quantitative basis (i.e., something like a factor of 3 or 5), and the possible reasons discussed (i.e., the primary emissions may be underestimated as has been found by McDonald et al., 2018).

Once a quantitative comparison is in hand, the authors should discuss how the comparison supports (or brings into question) the conclusions of the study.

2) The evaluation of the adjoint of the forward model through the comparisons illustrated in Figure 2 is quite compelling.

3) Pg. 9, lines 29-31: The statement given here seems to present a conundrum: "In other words, even though the population-weighted formaldehyde mixing ratio is larger in summer, scaling emissions uniformly year-round would result in a larger mixing ratio

change in winter." However, if emissions were reduced by 100%, then I would expect formaldehyde mixing ratios to be reduced to the small methane photochemical oxidation steady-state of 0.1 to 0.2 ppb (McCarthy et al., 2006) in both seasons. Does this imply that the secondary production of formaldehyde, which dominates in the summer, is very non-linear? Please fully discuss this apparent conundrum; a compelling explanation of the quoted statement is necessary for this result to be accepted as really representative of the atmosphere, and not an indication of a shortcoming of the modeling approach.

4) It would be useful to give an explanation for the cause of the result reported in the last sentence in Section 3.2. Why do formaldehyde mixing ratios increase when NOx emissions are decreased? Is this related to VOC sensitive-NOx saturated photochemical ozone formation? Is this possibly related to the conundrum mentioned in point 3 above?

5) Can Table 2 be expanded to give a complete budget for formaldehyde? Include additional columns for primary emissions, total sources and total losses. I would guess that the total sources and total losses should come close to balancing, with the difference due to net transport out of the modeling domain. This would then give the reader a useful summary of the relative magnitudes of sources and sinks of formaldehyde in the two seasons.

6) Pg. 11, lines 7: "Removal by dry deposition decreases by a factor of three between July and December." Other factors being equal, I expect dry deposition to simply scale with formaldehyde concentration, but this difference is larger than can be explained by the concentration differences in Table 1. Please explain how the "other factors" account for this finding.

7) Last paragraph, Section 3.3 – A much better description of exactly what is going here is required. I don't think that all "chemistry has been turned off", since that would turn off photochemical production of formaldehyde as well as photochemical loss processes, and result (I guess) in large negative sensitivities in summer. Please clarify.

8) Pg. 12, lines 3-5 state "Secondary formation of formaldehyde is enhanced by NO2 to NO conversions associated with the photolysis of NO2, and is inhibited by NO to NO2 conversions associated with the reaction of NO with ozone ...." This is an interesting finding, but the reason for and implications of this sensitivity is not discussed. Is this possibly related to point 4 above? Please discuss.

9) Pg. 12, lines 8-10 note "In Section 3.2, ISOPRENE, ETHENE, OLE1, and OLE2 were identified as the main contributors to secondary formaldehyde in July, yet none of the reactions that directly involve any of these species appear in Table 4." This is again an interesting observation, but it is not discussed further. What is the reason for this? Please discuss how this fits in the overall picture of formaldehyde sources and sinks that you all are developing.

10) The discussion in the last paragraph of Section 3.4 is not entirely correct. In particular the sensitivities of many of the reaction rates are similar between the two seasons. This discussion should be improved. Also, what is the cause of the opposite effects on formaldehyde of two competing fates for peroxy acetyl radicals ($CH_3C(=O)OO•$)? It seems many of my comments (3,4,8,10) all involve the role of NOx in formaldehyde formation. Can a coherent discussion of all of the NOx related issues be added in a separate section? In this regard, it would be useful to explicitly discuss the effect of reducing NOx emissions on population weighted formaldehyde exposure; if I understand correctly, reducing NOx emissions would worsen the formaldehyde health effects, but this is not explicitly discussed in the paper.

Minor issues:

1) Paragraph on Pg. 2 and 3 – In the discussion of the approaches used to estimate the relative importance of direct emissions versus secondary formation of formaldehyde, it may be useful to point out that the results for Houston are not expected to be similar to those from other urban areas (such as Mexico City) given the vast petrochemical

industrial activities in Houston.

2) Pg. 9, line 18: "Figures 3a and 3c" should be "Figures 3a and 3b" I guess.

3) Pg. 9, lines 28-29: I have a difficult time fully understanding and remembering the quantities discussed. I suggest that the sentence on these line be changed to more fully emphasize the quantities: "The air basin-wide response (Table 1) decreases by a factor of 1.6 between summer and winter, while its sensitivity to emissions (Figure 3) almost doubles."

References

McDonald, B.C., J.A. de Gouw, J.B. Gilman, S.H. Jathar, A. Akherati, C.D. Cappa, J.L. Jimenez, J. Lee-Taylor, P.L. Hayes, S.A. McKeen, Y.Y. Cui, S.-W. Kim, D.R. Gentner, G. Isaacman-VanWertz, A.H. Goldstein, R.A. Harley, G.J. Frost, J.M. Roberts, T.B. Ryerson, and M. Trainer, Volatile chemical products emerging as largest petrochemical source of urban organic emissions, Science, doi:10.1126/science.aaq0524, 2018.

Parrish, D.D., D.B. Millet, and A.H. Goldstein (2009), Increasing ozone in marine boundary layer air inflow at the west coasts of North America and Europe, Atmos. Chem. Phys., 9, 1303–1323.
* * *

---

## Author Comment (AC1) · 1 May 2019

The authors thank the Referees for their insightful comments which have helped us to improve the manuscript. In particular, performance of the forward model improved after values of inflow boundary conditions were revised.

Referees' comments and questions are formatted in ***bold italics***.

In our answers, modifications made to our original text are highlighted in red.

**Referee #1**

*Summary:*

[Figure]

*The paper presents an interesting analysis of formaldehyde sources in the San Francisco Bay area. This research is relevant because 1) formaldehyde is an air toxic to which significant health impacts have been attributed, and 2) the modeling methods presented here may well have wider applications. The study is well conceived and executed, and the paper is well written, with the procedures and results concisely and clearly explained. I recommend that the paper be published when the following comments are addressed. It is most important to address the major comments that question if everything is done correctly (major comments 1, 3, 6, 7) and less important to address the comments related to more discussion of formaldehyde sources and sinks.*

*Major issues and comments:*

*1) Section 2, which describes the model, the procedures for conducting the modeling, and the interpretation of the model results, is largely outside my area of expertise, so I can neither support nor critique most of that material. Section 3.1 provides the only evaluation of the model performance that is accessible to me, and I expect that this is true of most readers of the paper. The evaluation of the forward model performance is only through comparisons of monthly average diurnal cycles of mixing ratios of O3, NO and NO2 (Figures S2-S7) for the July and December simulation periods. However, the discussion is qualitative (e.g., "Agreement is good for O3 mixing ratios in July and December . . .."), and in my view, not entirely accurate. Some large errors in the O3 simulation are evident, so that I am not confident that the model simulation can be accurately used for its intended purpose. I recommend a more thoughtful evaluation be conducted that can provide some quantitative information regarding how well the model actually performed. In this regard, I have the following suggestions.*

*The three species O3, NO and NO2 are closely coupled in the atmosphere due to*

*rapid interconversion: NO + O3 ==> NO2 + O2; NO2 + hν + O2 ==> NO + O3. Hence, it makes sense to base the model-observation comparison on NOx = NO + NO2 and Ox = O3 + NO2 (which are not affected by the above interconversion) rather than on the separate species, whose concentrations are more difficult to interpret.*

*For O3 at any low elevation site in the Bay Area in the absence of continental (natural or anthropogenic) effects, I would expect a constant average diurnal profile at about 25 ppb in summer and 40 ppb in winter (see Parrish et al., 2009). The utility of the model is to simulate how concentrations differ from this zero-order expectation. To objectively evaluate the model performance, one should quantitatively compare how the simulated and observed Ox concentrations differ from this expectation. A qualitative examination of the figures suggests that the model may be substantially overestimating O3 (and therefore Ox) production in the SF Bay area in summer. If there is a quantitative overestimate in the simulated photochemical production of ozone, then the implications for the accuracy of the simulation of the photochemical production of formaldehyde must be discussed.*

*The observed NOx concentrations are expected to reflect anthropogenic emissions, so disagreements in a quantitative comparison of the simulated and observed NOx (or CO) concentrations may indicate errors in the emission inventory or in the simulation of vertical transport and horizontal advection. If the comparison is poor, then the reasons for the poor comparisons, and the implications for the conclusions of this paper must be discussed.*

*For the carbonyls, I cannot recommend an approach superior to that taken in Figures S8-S10, but the results can be more informatively summarized. Since one goal of this paper is to understand formaldehyde concentrations, any bias in the model simulations of the aldehydes should be quantitatively discussed. A*

*qualitative examination of the figures suggests that there is no significant bias in the formaldehyde results, but that acetaldehyde simulations may be biased high; these conclusions should be put on a quantitative basis. The model underestimate of acetone mixing ratios should also be put on a quantitative basis (i.e., something like a factor of 3 or 5), and the possible reasons discussed (i.e., the primary emissions may be underestimated as has been found by McDonald et al., 2018).*

*Once a quantitative comparison is in hand, the authors should discuss how the comparison supports (or brings into question) the conclusions of the study.*

These comments have led us to review some of our modeling inputs. In particular, we agree that boundary conditions that were used previously did not accurately represent inflow of ozone into the modeling domain. The algorithm used to create boundary conditions for our model from MOZART-4 outputs was updated accordingly to address this issue. New model simulations were run with updated boundary conditions, which led to significantly improved model performance. The manuscript has been revised to use results from these new simulations.

Furthermore, the model evaluation section now includes comparisons of observed versus modeled mixing ratios of $O_3 + NO_2$ and of $NO + NO_2$, as suggested by Referee #1. More generally, this section now includes a quantitative rather than qualitative description of the comparison between modeled and measured mixing ratios. Six figures and one table have been added to the supporting information to supplement the model evaluation.

*The forward model is evaluated by comparing modeled mixing ratios of a number of species to ground-based observations from routine monitoring stations (Fig. S1 through S16 and Table S1). Hourly measurements of $O_3$ were available at 20 sites, hourly measurements of NO, $NO_2$, and CO were available at 13 sites, and daily obser-*

*vations (24-hour averages) of formaldehyde, acetaldehyde, and acetone were available at 3 sites. The model captures the magnitude and diurnal variations of summer-time $O_3$ mixing ratios at a majority of the sites. The corresponding mean bias ranges from $-4.4$ to 10.5 ppb and is within $\pm$ 5 ppb at 12 out of the 20 locations where observations are available. The corresponding root mean square error (RMSE) ranges from 3.5 to 12.7 ppb. Comparing average daily profiles of $O_3 + NO_2$ mixing ratios yields mean bias in the range $-2.3$–7.0 ppb and RMSE in the range 2.3–7.6 ppb. Model performance is weaker for summer season $NO$, $NO_2$, and $NO_x = NO + NO_2$ when compared to results for $O_3$ and $O_3 + NO_2$. $NO_x$ is sometimes dramatically overestimated. In contrast, $CO$ is consistently underestimated by the model. This issue may be a consequence of the size of model grid cells: in the model, local emissions are instantaneously diluted within a 4 km x 4 km grid cell. This artificial dilution has more influence on pollutants that are directly emitted, such as $CO$ and $NO$, than on secondary pollutants such as ozone.*

*Model performance is less satisfactory in winter for ozone, oxides of nitrogen, and carbon monoxide (Table S1), which suggests that winter-time atmospheric chemistry is not as well represented by the model. However, results presented in the following sections indicate that atmospheric chemistry has little influence on formaldehyde concentrations in winter. In this case, formaldehyde behaves mostly as a non-reactive tracer species. The performance of the model to predict tracer species concentrations in both summer and winter has previously been evaluated by Bastien et al. (2015).*

*Here, the model evaluation includes comparisons of modeled versus observed daily-averaged mixing ratios of formaldehyde, acetaldehyde, and acetone (Fig. S14 through S16). Mean modeled mixing ratios of formaldehyde are within $\pm$ 26 % of observed values at all sites except Arkansas Street in San Francisco in July (relative difference = 34 %). There is no consistent trend (overestimation versus underestimation) in these differences. In contrast, mean mixing ratios of acetaldehyde are consistently underestimated by the model, but are within 34 % of observed values at all sites except Chapel Way in Fremont in July (relative difference = $-55$ %). The model consistently underes-*

*timates acetone mixing ratios by 56 to 81 %. A possible explanation for the underpredictions of acetone is that emissions may be significantly underestimated (McDonald et al., 2018).*

*2) The evaluation of the adjoint of the forward model through the comparisons illustrated in Figure 2 is quite compelling.*

Noted, thank you.

*3) Pg. 9, lines 29-31: The statement given here seems to present a conundrum: "In other words, even though the population-weighted formaldehyde mixing ratio is larger in summer, scaling emissions uniformly year-round would result in a larger mixing ratio change in winter." However, if emissions were reduced by 100%, then I would expect formaldehyde mixing ratios to be reduced to the small methane photochemical oxidation steady-state of 0.1 to 0.2 ppb (McCarthy et al., 2006) in both seasons. Does this imply that the secondary production of formaldehyde, which dominates in the summer, is very non-linear? Please fully discuss this apparent conundrum; a compelling explanation of the quoted statement is necessary for this result to be accepted as really representative of the atmosphere, and not an indication of a shortcoming of the modeling approach.*

If emissions were reduced by 100 %, formaldehyde mixing ratios would indeed be controlled by inflow of formaldehyde and by secondary formation from background methane and other precursors entering the modeling domain at the boundaries. This apparent conundrum does show that the relationship between emissions and formaldehyde mixing ratios in the model is non-linear. Whether this relationship is very non-linear is difficult to say. In any case, relying solely on first-order sensitivities to predict the behavior of an even moderately non-linear system is not recommended for pertur-

bations as large as 100 % of baseline values. This is now discussed in the revised manuscript:

*In other words, even though the population-weighted formaldehyde mixing ratio is larger in summer, scaling* *anthropogenic and/or biogenic* *emissions uniformly year-round would result in a larger mixing ratio change in winter.* *In practice, the effects of scaling emissions up or down should be limited to a range or perturbations where the relationship between emissions and formaldehyde mixing ratios is nearly linear. Although the boundaries of this linear range are not fully characterized, sensitivity coefficients reported here should not be applied to the case where all emissions are removed from the simulations (i.e., a 100 % perturbation). If primary emissions of formaldehyde and precursors are removed, formaldehyde mixing ratios should approach background values of a few tens of ppb in both summer and winter. Changes in model responses would in this case be larger in summer than in winter, which contradicts our findings. The use of first-order sensitivities to characterize the behavior of a non-linear system is nevertheless applicable to identifying emissions that should be prioritized in control strategies that aim to reduce ambient formaldehyde. To summarize, controlling anthropogenic emissions is less effective, to a first-order,* *at reducing mixing ratios when mixing ratios are higher (i.e. in summer).*

**4) It would be useful to give an explanation for the cause of the result reported in the last sentence in Section 3.2. Why do formaldehyde mixing ratios increase when NOx emissions are decreased? Is this related to VOC sensitive-NOx saturated photochemical ozone formation? Is this possibly related to the conundrum mentioned in point 3 above?**

In the revised manuscript, this discussion can be found in Section 3.4 (see our answer to comment #9 for more details):

*The sensitivity of formaldehyde responses to* $NO_x$ *emissions is negative in both seasons for all receptor areas, with a magnitude that ranges from 10 to 28 % of the corresponding overall sensitivity to emissions. As mentioned above, the negative sign of this sensitivity means that decreasing* $NO_x$ *emissions would increase formaldehyde concentrations and vice-versa. Causes of this negative sign are discussed in Section 3.4.*

**5) Can Table 2 be expanded to give a complete budget for formaldehyde? Include additional columns for primary emissions, total sources and total losses. I would guess that the total sources and total losses should come close to balancing, with the difference due to net transport out of the modeling domain. This would then give the reader a useful summary of the relative magnitudes of sources and sinks of formaldehyde in the two seasons.**

Columns were added to Table 2 as suggested. The following text was added to the manuscript:

*Table 2 shows total production and loss of formaldehyde in the modeling domain. The difference between these values is approximately equal to net transport out of the modeling domain. In winter, a larger fraction of the formaldehyde produced is transported out of the modeling domain rather than being lost due to chemical reactions or surface deposition. This trend is consistent with formaldehyde having a longer atmospheric lifetime in winter compared to summer.*

**6) Pg. 11, lines 7: "Removal by dry deposition decreases by a factor of three between July and December." Other factors being equal, I expect dry deposition to simply scale with formaldehyde concentration, but this difference is larger than can be explained by the concentration differences in Table 1. Please explain**

*how the "other factors" account for this finding.*

As requested, a more detailed explanation was added to the manuscript:

*Photochemical removal decreases by one to two orders of magnitude between July and December, and removal by dry deposition decreases by a factor of 2.5 between these two periods. Dry deposition fluxes are estimated as the product of surface-layer concentrations and deposition velocities. While deposition fluxes depend on atmospheric concentrations of relevant pollutants, there are also important seasonal effects on deposition velocities. Here, the decrease in dry deposition between seasons results from the combined effects of reduced formaldehyde concentrations (Table 1) and, in some areas, reduced vertical transport and mixing within the surface layer in winter compared to summer (Figure S20).*

*Reduced losses by dry deposition and by photochemical reactions both contribute to a longer atmospheric lifetime of formaldehyde in winter. This extended lifetime may in turn increase the sensitivity of model responses to directly emitted formaldehyde.*

Additionally, a new figure (Figure S20) was added to the supplementary material. This figure shows the dry deposition velocity ($v_d$), the surface-layer concentration ($C$), and the dry deposition flux ($F$) of formaldehyde, separately for the summer and winter simulation periods.

*7) Last paragraph, Section 3.3 – A much better description of exactly what is going here is required. I don't think that all "chemistry has been turned off", since that would turn off photochemical production of formaldehyde as well as photochemical loss processes, and result (I guess) in large negative sensitivities in summer. Please clarify.*

In this paragraph, we are trying to answer the question "can the reduction in photo-chemical removal of formaldehyde between summer and winter explain the 3-fold increase in the sensitivity of model responses to formaldehyde emissions between these two seasons?" The sensitivities we are referring to here are sensitivities of model responses to formaldehyde emissions:

$$\text{sensitivity} = \frac{\partial \; (\text{ population-weighted time-averaged ambient formaldehyde concentration })}{\partial \; (\text{ formaldehyde emissions })}$$

These sensitivities are positive because increases in formaldehyde emissions lead to higher ambient formaldehyde concentrations. When chemistry is turned off in the simulations, secondary formation and photochemical removal of formaldehyde are inactivated, but emissions are left unchanged. These sensitivities are still positive because more formaldehyde emissions still means higher ambient concentrations of formaldehyde. These sensitivities do increase and are still positive even when chemistry is turned off. The reason is that since photochemical removal is not occurring, the atmospheric lifetime of formaldehyde increases. As a consequence, each molecule of formaldehyde that is emitted is likely to remain in the modeling domain for a longer period of time, and has therefore increased opportunity to contribute to time-averaged concentrations.

More generally, in this section, we are trying to identify which parameters influence first-order sensitivities, which themselves characterize the influence of various parameters on model responses. This process is not unlike analyzing second-order sensitivities.

As the Referee suggests, we added a more detailed explanation of this point to the manuscript:

*The influence on sensitivities of photochemical removal of formaldehyde is investigated by running model simulations in which formaldehyde is treated as a tracer species: chemistry is turned off and there is neither secondary formation nor photochemical re-*

*moval of formaldehyde. In these simulations, removal of formaldehyde is due only to dry deposition and transport out of the modeling domain. The atmospheric lifetime of formaldehyde therefore increases. When the residence time of a particular molecule increases, on average these molecules stay in the modeling domain for longer periods of time and their concentrations increase. In other words, sensitivities of model responses to formaldehyde emissions are expected to be larger when chemistry is turned off. The goals of these additional simulations are to (i) characterize the influence of photochemical removal on sensitivities of model responses to formaldehyde emissions, and (ii) to investigate whether the large (∼200 %) increase of these sensitivities from July to December can be explained by the reduction of photochemical removal between these two seasons. With chemistry turned off in the model,* formaldehyde emissions sensitivities increased by 10 to 30 % in July, but by only a few percent in December (Table 3).

**8) Pg. 12, lines 3-5 state "Secondary formation of formaldehyde is enhanced by NO2 to NO conversions associated with the photolysis of NO2, and is inhibited by NO to NO2 conversions associated with the reaction of NO with ozone . . .." This is an interesting finding, but the reason for and implications of this sensitivity is not discussed. Is this possibly related to point 4 above? Please discuss.**

As the Referee suggests, this finding is discussed in greater detail in the revised manuscript (these additions also address comment #4):

*The* $OH$ *radical can remove formaldehyde from the atmosphere, but it can also oxidize other organic species to eventually form more formaldehyde. The latter effect prevails over the former, as indicated by the positive rate sensitivity of the* $OH$*-forming reaction involving* $H_2O$ *and* $O(^1D)$*.*

*[...]*

*We note the opposite effects on formaldehyde of two competing fates for peroxyacetyl radicals (*$CH_3C(O)OO^\bullet$*). The corresponding sensitivities are of comparable magnitudes but opposite signs. Reaction of these radicals with* $NO$ *forms* $CO_2$ *and methyl peroxy radicals (*$CH_3OO^\bullet$*). The latter are readily converted to formaldehyde. The corresponding rate sensitivity is, accordingly, positive. In contrast, reaction of peroxyacetyl radicals with* $NO_2$ *forms peroxyacetyl nitrate (PAN). PAN is relatively stable in winter when temperature is low. PAN can dissociate to release* $CH_3C(O)OO^\bullet$ *and* $NO_2$ *back to the atmosphere. As long as peroxyacetyl radicals and* $NO_2$ *are bound together as PAN, they cannot react with other species and contribute to formaldehyde production. The PAN-forming reaction therefore inhibits production of secondary formaldehyde, and the corresponding rate sensitivity is negative. More generally, secondary formation of formaldehyde is enhanced by photolysis of* $NO_2$*, whereas it is inhibited by* $NO$ *to* $NO_2$ *conversions associated with the reaction of* $NO$ *with ozone.* $NO$ *contributes to the formation of formaldehyde by converting a number of different peroxy radicals into alkoxy radicals which can be further converted to formaldehyde (Seinfeld and Pandis, 2016).* $NO_2$ *inhibits formation of formaldehyde by reacting with* $OH$ *radicals to form nitric acid and with peroxyacetyl radicals to form PAN. The removal of* $OH$ *radicals by the* $OH + NO_2$ *chain-terminating reaction is likely responsible for the negative sign of the sensitivities of formaldehyde to* $NO_x$ *emissions (Section 3.2).*

Table 4 now features the 16 highest-ranking reactions (i.e., four more compared to the previous version which listed twelve reactions).

**9) Pg. 12, lines 8-10 note "In Section 3.2, ISOPRENE, ETHENE, OLE1, and OLE2 were identified as the main contributors to secondary formaldehyde in July, yet none of the reactions that directly involve any of these species appear in Table 4." This is again an interesting observation, but it is not discussed further. What**

*is the reason for this? Please discuss how this fits in the overall picture of formaldehyde sources and sinks that you all are developing.*

This statement warrants further discussion. We added text to the manuscript:

*In Section 3.2,* *alkenes – more specifically ISOPRENE, ETHENE, OLE1, and OLE2 – were identified as the main precursors to secondary formaldehyde in July.*  *Yet, none of the reactions that directly involve any of these species appear in Table 4* *(the highest-ranking one among these reactions – reaction of ethene with OH – ranks $18^{th}$). These two seemingly contradictory observations suggest that these alkenes are reactive enough that the factor that limits formation of formaldehyde is not the rate at which the reactions occur (low sensitivity to reaction rates) but rather the availability of the alkenes as reactants (high sensitivity to emissions).*

We also added text to reduce the potential for confusion about the significance of the data presented in Table 4:

*It should be noted that ordering reactions by rate sensitivity versus ordering them by reaction rate yields different rankings.* *Rate sensitivities account for both the rate and the formaldehyde yield of the reaction. In other words, they account for how often the reaction occurs and how many molecules of formaldehyde are eventually formed or destroyed every time the reaction occurs. The rate coefficient sensitivities presented in Table 4 do not indicate, however, which reactions produce the most formaldehyde. Rather, they indicate how much formaldehyde concentrations would be affected if rate coefficients were to change.*

*10) The discussion in the last paragraph of Section 3.4 is not entirely correct. In particular the sensitivities of many of the reaction rates are similar between the*

*two seasons. This discussion should be improved.*

Only a few rate sensitivities are similar between the two seasons; most are smaller in winter. We rephrased the text in question as follows:

*As discussed previously, secondary formation of formaldehyde is far less significant in December than in July. This trend is also reflected in the high-ranking rate sensitivities, most of which are smaller in magnitude in December.*

*Also, what is the cause of the opposite effects on formaldehyde of two competing fates for peroxy acetyl radicals (CH3C(=O)OO)?*

See response to comment #8.

*It seems many of my comments (3,4,8,10) all involve the role of NOx in formaldehyde formation. Can a coherent discussion of all of the NOx related issues be added in a separate section? In this regard, it would be useful to explicitly discuss the effect of reducing NOx emissions on population weighted formaldehyde exposure; if I understand correctly, reducing NOx emissions would worsen the formaldehyde health effects, but this is not explicitly discussed in the paper.*

The revised manuscript mentions this finding in the text more explicitly:

*The sensitivity of formaldehyde responses to $NO_x$ emissions is negative in both seasons for all receptor areas, with a magnitude that ranges from 10 to 28 % of the corresponding overall sensitivity to emissions. The negative sign of this sensitivity means that decreasing $NO_x$ emissions would increase formaldehyde concentrations at receptors and vice-versa. Causes of this negative sign are discussed in Section 3.4.*

*Minor issues:*

*1) Paragraph on Pg. 2 and 3 – In the discussion of the approaches used to estimate the relative importance of direct emissions versus secondary formation of formaldehyde, it may be useful to point out that the results for Houston are not expected to be similar to those from other urban areas (such as Mexico City) given the vast petrochemical industrial activities in Houston.*

We added the following to the revised manuscript:

*Regardless of the method being used, results derived for one location may not be directly applicable to another because several factors, including emissions and meteorology, can influence secondary formation of formaldehyde and of other species. For example, many petrochemical industries are located in Houston and its surroundings. Emissions from this area may be particularly different from those of other large cities.*

*2) Pg. 9, line 18: "Figures 3a and 3c" should be "Figures 3a and 3b" I guess.*

This has been corrected.

*3) Pg. 9, lines 28-29: I have a difficult time fully understanding and remembering the quantities discussed. I suggest that the sentence on these line be changed to more fully emphasize the quantities: "The air basin-wide response (Table 1) decreases by a factor of 1.6 between summer and winter, while its sensitivity to emissions (Figure 3) almost doubles."*

This text has been revised as follows:

*The air basin-wide response (R) decreases by a factor of 1.4 between summer and*

*winter, while its sensitivity to emissions $(\partial R/\partial E)$ almost doubles.*

**Referee #2**

*General comments:*

**The manuscript is very well structured and written. The manuscript is very interesting, adding more information on the use of the adjoint technique, in this case to calculate the sensitivity of formaldehyde to emissions of NOx, formaldehyde, and VOC precursors. I think the article deserves publication. My only concern is related to the linearity approach as I explain in my comments below.**

*Specific comments:*

*1. Please identify the radicals in R1 and R3 with more conventional radical identifiers.*

This has been done.

*2. Pg 6. "These studies suggest that ozone responses vary nearly linearly with emissions of precursor species for perturbations up to about 15–20 % of the nominal value." Could the authors discuss this a bit more? Does this apply also for NOx emissions?*

As suggested, this topic is now discussed in greater detail in the revised manuscript:

*Previous modeling studies investigated the range of perturbations over which Equation 7 holds reasonably well. Vuilleumier et al. (1997) used a Green's function approach in a box model to calculate first- and second-order sensitivities of ozone to initial concen-*

[Figure]

*trations of precursors and to reaction rate coefficients. They found that using first-order sensitivities to estimate changes in modeled ozone resulting from perturbing model parameters yielded acceptable results for perturbations of at most 25 % of the baseline values. Hakami et al. (2003, 2004) found good agreement ($r^2 \geqslant 0.99$) between first-order sensitivities of ozone to domain-wide NO and VOC emissions calculated using the DDM on the one-hand and a central finite-difference approximation with $\pm$ 10 % perturbations on the other hand. Additionally, Hakami et al. found that modeled ozone exhibited more nonlinear behavior when (i) perturbing $NO_x$ emissions compared to perturbing VOC emissions; (ii) ozone concentrations were high; and (iii) ozone chemistry was in the transition regime between VOC- and $NO_x$-controlled conditions. Cohan et al. (2005) found that changes in modeled ozone resulting from 10 % perturbations in domain-wide $NO_x$ and VOC emissions were accurately estimated using a Taylor series truncated after the first-order term. Overall, these studies suggest that ozone responses vary nearly linearly with emissions of precursor species for perturbations up to about 10 % of the nominal value, and up to 25 % in some cases. Since formaldehyde is a ubiquitous intermediate species in atmospheric oxidation cycles, we hypothesize that formaldehyde responses also vary approximately linearly with precursor emissions for such perturbations. Using the sensitivity results presented in this study for large (> 25 %) perturbations that are likely to result in nonlinear behavior should be avoided.*

**3. Pg. 6. The authors assume that (for formaldehyde) the "responses investigated here also vary approximately linearly with precursor emissions for perturbations up to about 15–20 %". Even considering a potential linearity in O3 response in the suggested range, the chemistry of formaldehyde is different. To support this second approach (similarity to O3), could the authors include (if possible and available) more discussion, based on other studies, chambers, etc.?**

The evaluation of adjoint sensitivities that we conduct in the manuscript (Fig. 2) does provide information on the non-linearity of the formaldehyde response when emissions of precursors are perturbed. We added text that discusses this topic:

*The good agreement between brute-force and adjoint sensitivities of formaldehyde to emissions of precursors other than NO supports the hypothesis that formaldehyde responses vary nearly linearly with these emissions for perturbations on the order of 10 % of baseline values; the brute-force approach includes non-linearities whereas first-order adjoint sensitivities do not. This hypothesis is likely to hold for somewhat larger perturbations since the perturbations used for creating Fig. 2 were in most cases larger than 10 % of baseline values. A likely explanation for the larger mismatch for NO shown in Fig. 2 is that non-linearities in the relationship between model responses and emissions are stronger for $NO_x$ than for emissions of other species.*

 *4. S 3.1. l-30 "The agreement is less satisfactory for perturbations in NO emissions". This is in line with my previous comment, on the hypothesis of the linearity of the formaldehyde response to precursor emissions. The comparison with the brute force approach indicates that this assumption of linearity is not completely valid. The conclusion section should include some comments on this uncertainty, to help the reader to interpret these results and to figure out the aspects that could be affected by this assumption of linearity.*

A paragraph was added to the conclusion to discuss limitations related to this issue:

*This study uses first-order sensitivity coefficients to characterize how formaldehyde at different receptors responds to changes in emissions and reaction rate coefficients. The use of first-order sensitivity coefficients to estimate changes in responses of non-linear systems (Eq. 8 and 9) while ignoring higher-order terms of the Taylor series*

*(Eq. 5) is subject to restrictions. The changes in parameters must be relatively small so that the system response remains approximately linear. The first-order sensitivities presented here should therefore not be used to estimate the effects on ambient formaldehyde of very large emission changes such as 100 % reductions. Perturbing NO emissions was found to yield particularly non-linear behavior. Additionally, first-order sensitivities should be recalculated if emissions or other model parameters were to change significantly from the baseline values used here. This study nevertheless provides a basis to identifying the emissions and chemical reactions that have the most influence on ambient formaldehyde.*

**Other revisions**

Revisions to the manuscript are described above in our responses to the Referees' comments and questions. Additionally, the following changes have been made:

- $CH_3C(=O)OO^\bullet$ was re-written as $CH_3C(O)OO^\bullet$ (without the equal sign).

- Following the new model simulations with updated boundary conditions (see details above), numerical values have been updated throughout the manuscript. The conclusions of the study remain unchanged.

- Other minor modifications have been made to the text.

- The model source code is now included in the supplementary material.